# An Improved Analysis of Gradient Tracking for Decentralized Machine Learning

**Anastasia Koloskova**
EPFL
anastasia.koloskova@epfl.ch

**Tao Lin**
EPFL
tao.lin@epfl.ch

**Sebastian U. Stich**
EPFL*
sebastian.stich@epfl.ch

## Abstract

We consider decentralized machine learning over a network where the training data is distributed across $n$ agents, each of which can compute stochastic model updates on their local data. The agent's common goal is to find a model that minimizes the average of all local loss functions. While gradient tracking (GT) algorithms can overcome a key challenge, namely accounting for differences between workers' local data distributions, the known convergence rates for GT algorithms are not optimal with respect to their dependence on the mixing parameter $p$ (related to the spectral gap of the connectivity matrix).

We provide a tighter analysis of the GT method in the stochastic strongly convex, convex and non-convex settings. We improve the dependency on $p$ from $\mathcal{O}(p^{-2})$ to $\mathcal{O}(p^{-1}c^{-1})$ in the noiseless case and from $\mathcal{O}(p^{-3/2})$ to $\mathcal{O}(p^{-1/2}c^{-1})$ in the general stochastic case, where $c \geq p$ is related to the negative eigenvalues of the connectivity matrix (and is a constant in most practical applications). This improvement was possible due to a new proof technique which could be of independent interest.

## 1 Introduction

Methods that train machine learning models on decentralized data offer many advantages over traditional centralized approaches in core aspects such as data ownership, privacy, fault tolerance and scalability [12, 33]. Many current efforts in this direction come under the banner of federated learning [17, 29, 28, 12], where a central entity orchestrates the training and collects aggregate updates from the participating devices. Fully decentralized methods, that do not rely on a central coordinator and that communicate only with neighbors in an arbitrary communication topology, are still in their infancy [24, 18].

The work of Lian et al. [24] on decentralized stochastic gradient descent (D-SGD) has spurred the research on decentralized training methods for machine learning models. This lead to improved theoretical analyses [16] and to improved practical schemes, such as support for time-varying topologies [32, 3, 16] and methods with communication compression [45, 51, 15, 47]. One of the most challenging aspect when training over decentralized data is data-heterogeneity, i.e. training data that is in a non-IID fashion distributed over the devices (for instance in data-center training) or generated in non-IID fashion on client devices [21, 13, 22, 23]. For example, the D-SGD method has been shown to be affected by the heterogenity [16].

In contrast, certain methods can mitigate the impact of heterogeneous data in decentralized optimization. For instance the *gradient tracking* (GT) methods developed by Lorenzo and Scutari [26] and Nedić et al. [34], or the later D$^2$ method by Tang et al. [46] which is designed for communication typologies that remain fixed and do not change over time.

---

*Current affiliation: CISPA Helmholtz Center for Information Security.

35th Conference on Neural Information Processing Systems (NeurIPS 2021).

Table 1: Important advances for Gradient Tracking in the strongly convex case. Our analysis improves upon all prior rates for both with and without the stochastic noise in terms of the graph parameter $p$.

| Reference | rate of convergence to $\epsilon$-accuracy | considered stochastic noise |
|---|---|---|
| Nedić et al. [34] | $\mathcal{O}\left(\dfrac{L^3}{\mu^3 p^2}\log\frac{1}{\varepsilon}\right)$ | ✗ |
| Alghunaim et al. [1] | $\mathcal{O}\left(\dfrac{L}{\mu}\log\frac{1}{\varepsilon}+\dfrac{1}{p^2}\log\frac{1}{\varepsilon}\right)$ | ✗ |
| Qu and Li [40] | $\mathcal{O}\left(\dfrac{L^2}{\mu^2 p^2}\log\frac{1}{\varepsilon}\right)$ | ✗ |
| Pu and Nedić [39] | $\tilde{\mathcal{O}}\left(\dfrac{\sigma^2}{\mu n\varepsilon}+\dfrac{\sqrt{L}\sigma}{\mu\sqrt{p}p\sqrt{\varepsilon}}+\dfrac{C_1}{\sqrt{\varepsilon}}\right)^a$ | ✓ |
| this work | $\tilde{\mathcal{O}}\left(\dfrac{\sigma^2}{\mu n\varepsilon}+\dfrac{\sqrt{L}\sigma}{\mu\sqrt{p}c\sqrt{\varepsilon}}+\dfrac{L}{\mu pc}\log\frac{1}{\varepsilon}\right)$ | ✓ |

---

$^a C_1$ is a constant that is independent of $\varepsilon$, but can depend on other parameters, such as $\sigma, \mu, L, p$

It is well known that GT methods do not depend on the heterogeneity of the data and that they converge linearly on distributed strongly convex problem instances without stochastic noise [26, 34]. However, when we apply these methods in the context of machine learning, we need to understand how they are affected by stochastic noise and how they behave on non-convex tasks.

In this paper, we develop a new, and improved, analysis of the gradient tracking algorithm with a novel proof technique. Along with the parallel contribution [55] that developed a tighter analysis of the $D^2$ algorithm, we now have a more accurate understanding of in which setting GT works well and in which ones it does not, and our results allow for a more detailed comparison between the D-SGD, GT and $D^2$ methods (see Section 5 below).

Our analysis improves over all existing results that analyze the GT algorithm. Specifically, we prove a weaker dependence on the connectivity of the network (spectral gap) which is commonly incorporated into the convergence rates via the standard parameter $p$. For example, in the strongly convex setting with stochastic noise we prove that GT converges at the rate $\tilde{\mathcal{O}}\big(\frac{\sigma^2}{n\varepsilon}+\frac{1}{c}\cdot\big(\frac{\sigma}{\sqrt{p\varepsilon}}+\frac{1}{p}\log\frac{1}{\varepsilon}\big)\big)$ where $\sigma^2$ is an upper bound on the variance of the stochastic noise, and $c\geq p$ a new parameter (often a constant). By comparing this result with the previously best known upper bound, $\tilde{\mathcal{O}}\big(\frac{\sigma^2}{n\varepsilon}+\frac{1}{p}\cdot\big(\frac{\sigma}{\sqrt{p\varepsilon}}+\frac{1}{p}\log\frac{1}{\varepsilon}\big)\big)$, by Pu and Nedić [39], we see that our upper bound improves the last two terms by a factor of $\frac{c}{p}\geq 1$ and that the first term matches with known lower bounds [37]. The $D^2$ algorithm [46] only converges under the assumption that $c$ is a constant[2] and the recent upper bound from [55] coincides with our worst case complexity for GT on all topologies where $D^2$ can be applied. We provide additional comparison of GT convergence rates in the Tables 1 and 2.

**Contributions.** Our main contributions can be summarized as:

- We prove better complexity estimates for the GT algorithm than known before with a new proof technique (which might be of independent interest).
- In the non-asymptotic regime (of importance in practice), the convergence rate depends on the network topology. By defining new graph parameters, we can give a tighter description of this dependency, explaining why the worst case behavior is rarely observed in practice (see Section 5.1). We verify this dependence in numerical experiments.
- We show that in the presence of stochastic noise, the leading term in the convergence rate of GT is optimal—we are the first to derive this in the non-convex setting—and matching the unimprovable rate of all-reduce mini-batch SGD.

## 2 Related Work

**Decentralized Optimization.** Decentralized optimization methods have been studied for decades in the optimization and control community [48, 30, 52, 6]. Many decentralized optimization methods

---

$^2$In $D^2$ the smallest eigenvalue of the mixing matrix $W$ must bounded from below: $\min_i \lambda_i(W)\geq -\frac{1}{3}$.

Table 2: Important advances for Gradient Tracking in the non-convex case. Our result improves upon all existing rates in terms of the graph parameter $p$.

| Reference | rate of convergence to $\epsilon$-accuracy | considered stochastic noise |
|---|---|---|
| Lorenzo and Scutari [26] | asymptotic convergence guarantees | ✗ |
| Zhang and You [60] | $\mathcal{O}\left(\dfrac{Ln\sigma^2}{\varepsilon^2} + \dfrac{Ln}{p^3\varepsilon}\right)$ | ✓ |
| Lu et al. [27] | $\mathcal{O}\left(\dfrac{C_1 + C_2\sigma}{\varepsilon^2}\right)^a$ | ✓ |
| this work | $\tilde{\mathcal{O}}\left(\dfrac{L\sigma^2}{n\varepsilon^2} + \dfrac{L\sigma}{(\sqrt{p}c + p\sqrt{n})\varepsilon^{3/2}} + \dfrac{L}{pc\varepsilon}\right)$ | ✓ |

$^a C_1$ and $C_2$ are constants that are independent of $\varepsilon$, but can depend on other parameters, such as $\sigma, n, L, p$.

[30, 11] are based on gossip averaging [14, 53, 4]. Such methods usually also work well on non-convex problems and can be used used for training deep neural networks [3, 24, 46]. There exists other methods, such as based on alternating direction method of multipliers (ADMM) [52, 10], dual averaging [6, 31, 41], primal-dual methods [2, 19], block-coordinate methods for generalized linear models [8] or using new gradient propagation mechanisms [50].

**Decentralized Optimization with Heterogeneous Objective Functions.** There exists several algorithms that are agnostic to data-heterogeneity. Notably, EXTRA [42] and decentralized primal-dual gradient methods [2] do not depend on the data heterogeneity and achieve linear convergence in the strongly convex noiseless setting. However, these algorithms are not designed to be used for non-convex tasks.

$D^2$ [46, 55] (also known as exact diffusion [56, 57]) and Gradient Tracking (GT) [26] (also known as NEXT [26] or DIGing [34]) are both algorithms that are agnostic to the data heterogeneity level, can tolerate the stochastic noise, and that can be applied to non-convex objectives such as the training of deep neural networks in machine learning. A limitation of the $D^2$ algorithm is that it is not clear how it can be applied to time-varying topologies, and that it can only be used on constant mixing topologies with negative eigenvalue bounded from below by $-\frac{1}{3}$. Other authors proposed algorithms that perform well on heterogeneous DL tasks [25, 59], but theoretical proofs that these algorithms are independent of the degree of heterogeneity are still pending.

**Gradient Tracking.** There is a vast literature on the Gradient Tracking method itself. A tracking mechanism was used by Zhu and Martínez [61] as a way to track the average of a distributed continuous process. Lorenzo and Scutari [26] applied this technique to track the gradients, and analyzed its asymptotic behavior in the non-convex setting with a time-varying topologies. Nedić et al. [34] analyze GT (named as DIGing) in the strongly convex noiseless case with a time-varying network. Qu and Li [40] extend the GT analysis to the non-convex, weakly-convex and strongly convex case without stochastic noise. Nedić et al. [35] allow the different stepsizes on different workers. Yuan et al. [58] analyze asymptotic behavior of GT for dynamic optimization. Pu and Nedić [39] studied the GT method on stochastic problems and strongly convex objectives. Further, Xin et al. [54] analyze asymptotic behavior of GT with stochastic noise. For non-convex stochastic functions GT was analyzed by Zhang and You [60] and Lu et al. [27]. Li et al. [20] combine GT with variance reduction to achieve linear convergence in the stochastic case. Tziotis et al. [49] obtain second order guarantees for GT.

## 3 Setup

We consider optimization problems where the objective function is distributed across $n$ nodes,

$$\min_{\mathbf{x}\in\mathbb{R}^d}\left[f(\mathbf{x}) := \frac{1}{n}\sum_{i=1}^{n}\left[f_i(\mathbf{x}) = \mathbb{E}_{\xi\sim\mathcal{D}_i}F_i(\mathbf{x},\xi)\right]\right], \tag{1}$$

where $f_i\colon \mathbb{R}^d \to \mathbb{R}$ denotes the local function available to the node $i$, $i \in [n] := \{1,\dots n\}$. Each $f_i$ is a stochastic function $f_i(\mathbf{x}) = \mathbb{E}_{\xi\sim\mathcal{D}_i}F_i(\mathbf{x},\xi)$ with access only to stochastic gradients $\nabla F_i(\mathbf{x},\xi)$. This setting covers empirical risk minimization problems with $\mathcal{D}_i$ being a uniform distribution over the local training dataset. It also covers deterministic optimization when $F_i(\mathbf{x},\xi) = f_i(\mathbf{x}), \forall\xi$.

We consider optimization over a decentralized network, i.e. when there is an underlying communication graph $G = (V, E)$, $|V| = n$, each of the nodes (e.g. a connected device) can communicate only along the edges $E$. In decentralized optimization it is convenient to parameterize communication by a mixing matrix $W \in \mathbb{R}^{n \times n}$, where $w_{ij} = 0$ if and only if nodes $i$ and $j$ are not communicating, $(i, j) \notin E$.

**Definition 1** (Mixing Matrix). *A matrix with non-negative entries $W \in [0, 1]^{n \times n}$ that is symmetric ($W = W^\top$) and doubly stochastic ($W\mathbf{1} = \mathbf{1}$, $\mathbf{1}^\top W = \mathbf{1}^\top$), where $\mathbf{1}$ denotes the all-one vector in $\mathbb{R}^n$.*

### 3.1 Notation

We use the notation $\mathbf{x}_i^{(t)} \in \mathbb{R}^d$, $\mathbf{y}_i^{(t)} \in \mathbb{R}^d$ to denote the iterates and the tracking sequence, respectively, on node $i$ at time step $t$. For vectors $\mathbf{z}_i \in \mathbb{R}^d$ ($\mathbf{z}_i$ could for instance be $\mathbf{x}_i^{(t)}$ or $\mathbf{y}_i^{(t)}$) defined for $i \in [n]$ we denote by $\bar{\mathbf{z}} = \frac{1}{n} \sum_{i=1}^{n} \mathbf{z}_i$.

We use both vector and matrix notation whenever it is more convenient. For vectors $\mathbf{z}_i \in \mathbb{R}^d$ defined for $i \in [n]$ we denote by a capital letter the matrix with columns $\mathbf{z}_i$, formally

$$Z := [\mathbf{z}_1, \ldots, \mathbf{z}_n] \in \mathbb{R}^{d \times n}, \qquad \bar{Z} := [\bar{\mathbf{z}}, \ldots, \bar{\mathbf{z}}] \equiv Z\tfrac{1}{n}\mathbf{1}\mathbf{1}^\top, \qquad \Delta Z := Z - \bar{Z}. \quad (2)$$

We extend this definition to gradients of (1), with $\nabla F(X^{(t)}, \xi^{(t)}), \nabla f(X^{(t)}) \in \mathbb{R}^{d \times n}$:

$$\nabla F(X^{(t)}, \xi^{(t)}) := \left[ \nabla F_1(\mathbf{x}_1^{(t)}, \xi_1^{(t)}), \ldots, \nabla F_n(\mathbf{x}_n^{(t)}, \xi_n^{(t)}) \right],$$

$$\nabla f(X^{(t)}) := \left[ \nabla f(\mathbf{x}_1^{(t)}), \ldots, \nabla f(\mathbf{x}_n^{(t)}) \right].$$

### 3.2 Algorithm

The Gradient Tracking algorithm (or NEXT, DIGing) can be written as

$$\begin{pmatrix} X^{(t+1)} \\ \gamma Y^{(t+1)} \end{pmatrix}^\top = \begin{pmatrix} X^{(t)} \\ \gamma Y^{(t)} \end{pmatrix}^\top \begin{pmatrix} W & 0 \\ -W & W \end{pmatrix} + \gamma \begin{pmatrix} 0 \\ \nabla F(X^{(t+1)}, \xi^{(t+1)}) - \nabla F(X^{(t)}, \xi^{(t)}) \end{pmatrix}^\top \quad \text{(GT)}$$

in matrix notation. Here and $X^{(t)} \in \mathbb{R}^{d \times n}$ denotes the iterates, $Y^{(t)} \in \mathbb{R}^{d \times n}$, with $Y^{(0)} = \nabla F(X^{(t)}, \xi^{(t)})$ the sequence of tracking variables, and $\gamma > 0$ denotes the stepsize. This update is summarized in Algorithm 1.

---

**Algorithm 1** GRADIENT TRACKING

---

**input** Initial values $\mathbf{x}_i^{(0)} \in \mathbb{R}^d$ on each node $i \in [n]$, communication graph $G = ([n], E)$ and mixing matrix $W$, stepsize $\gamma$, initialize $\mathbf{y}_i^{(0)} = \nabla F_i(\mathbf{x}_i^{(0)}, \xi_i^{(0)})$, $\mathbf{g}_i^{(0)} = \mathbf{y}_i^{(0)}$ in parallel for $i \in [n]$.

1: **in parallel on all workers** $i \in [n]$, **for** $t = 0, \ldots, T - 1$ **do**

2:     each node $i$ sends $\left( \mathbf{x}_i^{(t)}, \mathbf{y}_i^{(t)} \right)$ to is neighbors

3:     $\mathbf{x}_i^{(t+1)} = \sum_{j:\{i,j\} \in E} w_{ij} \left( \mathbf{x}_j^{(t)} - \gamma \mathbf{y}_j^{(t)} \right)$                    ▷ update model parameters

4:     Sample $\xi_i^{(t+1)}$, compute gradient $\mathbf{g}_i^{(t+1)} = \nabla F_i \left( \mathbf{x}_i^{(t+1)}, \xi_i^{(t+1)} \right)$

5:     $\mathbf{y}_i^{(t+1)} = \sum_{j:\{i,j\} \in E} w_{ij} \mathbf{y}_j^{(t)} + \left( \mathbf{g}_i^{(t+1)} - \mathbf{g}_i^{(t)} \right)$                    ▷ update tracking variable

6: **end parallel for**

---

Each node $i$ stores and updates two variables, the model parameter $\mathbf{x}_i^{(t)}$ and the tracking variable $\mathbf{y}_i^{(t)}$. The model parameters are updated on line 3 with a decentralized SGD update but using $\mathbf{y}_i^{(t)}$ instead of a gradient. Variable $\mathbf{y}_i^{(t)}$ tracks the average of all local gradients on line 5. Intuitively, the algorithm is agnostic to the functions heterogeneity because $\mathbf{y}_i^{(t)}$ is 'close' to the full gradient of $f(\mathbf{x})$ (suppose we would replace line 5 with exact averaging in every timestep, then $\mathbf{y}_i^{(t+1)} = \frac{1}{n} \sum_{i=1}^{n} \mathbf{g}_i^{(t+1)}$. For further discussion of the tracking mechanism refer to [26, 34, 39].

| graph/topology | $1/p$ | $c$ |
|---|---|---|
| ring | $\mathcal{O}(n^2)$ | $8/9$ |
| 2d-torus | $\mathcal{O}(n)$ | $\geq 4/5$ |
| fully connected | $\mathcal{O}(1)$ | $1$ |

Table 3: Parameters $p$ and $c$ for some common network topologies on $n$ nodes for uniformly averaging $W$, i.e. $w_{ij} = \frac{1}{deg(i)} = \frac{1}{deg(j)}$ for $\{i, j\} \in E$, see e.g. [36].

### 3.3 Assumptions

We first state an assumption on the mixing matrix.

**Assumption 1** (Mixing Matrix). *Let $\lambda_i(W)$, $i \in [n]$, denote the eigenvalues of the mixing matrix $W$ with $1 = \lambda_1(W) > \lambda_2(W) \geq \cdots \geq \lambda_n(W) > -1$. With this, we can define the spectral gap $\delta = 1 - \max\{|\lambda_2(W)|, |\lambda_n(W)|\}$, and the mixing parameters*

$$p = 1 - \max\{|\lambda_2(W)|, |\lambda_n(W)|\}^2, \qquad c = 1 - \min\{\lambda_n(W), 0\}^2. \qquad (3)$$

*We assume that $p > 0$ (and consequently $c > 0$).*

The assumption $p > 0$ ensures that the network topology is connected, and that the consensus distance decreases linearly after each averaging step, i.e. $\left\| XW - \bar{X} \right\|_F^2 \leq (1-p) \left\| X - \bar{X} \right\|_F^2$, $\forall X \in \mathbb{R}^{d \times n}$. The parameter $p$ is closely related to the spectral gap $\delta$ as it holds $p = 2\delta - \delta^2$. From this we can conclude that $\delta \leq p \leq 2\delta$ and, asymptotically for $\delta \to 0$, $p \to 2\delta$. Assuming a lower bound on $p$ (or equivalently $\delta$) is a standard assumption in the literature.

The parameter $c$ is related to the most negative eigenvalue. From the definition (3) it follows that the auxiliary mixing parameter $c \geq p$ for all mixing matrices $W$. The parameters $p$ and $c$ are only equal when $|\lambda_n(W)| \geq |\lambda_2(W)|$ and $\lambda_n(W) \leq 0$. Moreover, if the diagonal entries $w_{ii}$ (self-weights) of the mixing matrix are all strictly positive, then $c$ has to be strictly positive.

**Remark 1** (Lower bound on $c$.). *Let $W$ be a mixing matrix with diagonal entries (self-weights) $w_{ii} \geq \rho > 0$, for a parameter $\rho$. Then $\lambda_n(W) \geq 2\rho - 1$ and $c \geq \min\{2\rho, 1\}$.*

This follows from Gershgorin's circle theorem [7] that guarantees $\lambda_n(W) \geq 2\rho - 1$, and hence $c \geq 1 - \min\{2\rho - 1, 0\}^2 \geq \min\{2\rho, 1\}$.

For many choices of $W$ considered in practice, most notably when the graph $G$ has constant node-degree and the weights $w_{ij}$ are chosen by the popular Metropolis-Hastings rule, i.e. $w_{ij} = w_{ji} = \min\{\frac{1}{\deg(i)+1}, \frac{1}{\deg(j)+1}\}$ for $(i, j) \in E$, $w_{ii} = 1 - \sum_{j=1}^n w_{ij} \geq \frac{1}{\max_{j \in [n]} \deg(j)}$, see also [53, 4]. In this case, the parameter $c$ can be bounded by a constant depending on the maximal degree. Moreover, for any given $W$, considering $\frac{1}{2}(W + I_n)$ instead (i.e. increasing the self-weights), ensures that $c = 1$. However, in contrast to e.g. the analysis in [55] we do not need to pose an explicit bound on $c$ as an assumption. In practice, for many graphs, the parameter $c$ is bounded by a constant (see Table 3).

We further use the following standard assumptions:

**Assumption 2** (*L*-smoothness). *Each function $f_i \colon \mathbb{R}^d \to \mathbb{R}$, $i \in [n]$ is differentiable and there exists a constant $L \geq 0$ such that for each $\mathbf{x}, \mathbf{y} \in \mathbb{R}^d$:*

$$\|\nabla f_i(\mathbf{y}) - \nabla f_i(\mathbf{x})\| \leq L \|\mathbf{x} - \mathbf{y}\|. \qquad (4)$$

Sometimes we will in addition assume that the functions are (strongly) convex.

**Assumption 3** ($\mu$-strong convexity). *Each function $f_i \colon \mathbb{R}^d \to \mathbb{R}$, $i \in [n]$ is $\mu$-strongly convex for constant $\mu \geq 0$, i.e. for all $\mathbf{x}, \mathbf{y} \in \mathbb{R}^d$:*

$$f_i(\mathbf{x}) - f_i(\mathbf{y}) + \frac{\mu}{2} \|\mathbf{x} - \mathbf{y}\|_2^2 \leq \langle \nabla f_i(\mathbf{x}), \mathbf{x} - \mathbf{y} \rangle. \qquad (5)$$

**Assumption 4** (Bounded noise). *We assume that there exists constant $\sigma$ s.t. $\forall \mathbf{x}_1, \dots \mathbf{x}_n \in \mathbb{R}^d$*

$$\frac{1}{n} \sum_{i=1}^n \mathbb{E}_{\xi_i} \|\nabla F_i(\mathbf{x}_i, \xi_i) - \nabla f_i(\mathbf{x}_i)\|_2^2 \leq \sigma^2. \qquad (6)$$

We discuss possible relaxations of these assumptions in Section 4.1 below.

# 4 Convergence results

We now present our novel convergence results for GT in Section 4.1 and Section 4.2 below. We provide a proof sketch to explain the key difficulties and technical novelty compared to prior results later in the next Section 6.

## 4.1 Main theorem—GT convergence in the general case

**Theorem 2.** *Let $\mathbf{x}_i^{(t)}$, $i \in [n]$, $T > \frac{2}{p} \log \left( \frac{50}{p} (1 + \log \frac{1}{p}) \right)$ denote the iterates of the GT Algorithm 1 with a mixing matrix as in Definition 1. If Assumptions 1, 2 and 4 hold, then there exists a stepsize $\gamma$ such that the optimization error is bounded as follows:*
***Non-convex:*** *Let $F_0 = f(\bar{\mathbf{x}}^{(0)}) - f^\star$ for $f^\star \leq \min_{\mathbf{x} \in \mathbb{R}^d} f(\mathbf{x})$. Then it holds*

$$\frac{1}{T+1} \sum_{t=0}^{T} \left\| \nabla f(\bar{\mathbf{x}}^{(t)}) \right\|_2^2 \leq \varepsilon \,, \text{ after } \tilde{\mathcal{O}} \left( \frac{\sigma^2}{n\varepsilon} + \frac{\sigma}{(\sqrt{p}c + p\sqrt{n})\varepsilon^{3/2}} + \frac{1 + L\tilde{R}_0^2 F_0^{-1}}{pc\varepsilon} \right) \cdot LF_0 \text{ iterations.}$$

***Strongly-convex:*** *Under the additional Assumption 3 with $\mu > 0$ and weights $w_t \geq 0$, $W_T = \sum_{t=0}^{T} w_t$, specified in the proof, it holds for $R_{T+1}^2 = \left\| \bar{\mathbf{x}}^{(T+1)} - \mathbf{x}^\star \right\|^2$:*

$$\sum_{t=0}^{T} \frac{w_t}{W_T} \left[ \mathbb{E} f(\bar{\mathbf{x}}^{(t)}) - f^\star \right] + \frac{\mu}{2} R_{T+1} \leq \varepsilon \,, \text{ after } \tilde{\mathcal{O}} \left( \frac{\sigma^2}{\mu n \varepsilon} + \frac{\sqrt{L}\sigma}{\mu \sqrt{p}c\sqrt{\varepsilon}} + \frac{L}{\mu pc} \log \frac{1}{\varepsilon} \right) \text{ iterations.}$$

***General convex:*** *Under the additional Assumption 3 with $\mu \geq 0$, it holds for $R_0^2 = \left\| \bar{\mathbf{x}}^{(0)} - \mathbf{x}^\star \right\|^2$:*

$$\frac{1}{T+1} \sum_{t=0}^{T} \left[ \mathbb{E} f(\bar{\mathbf{x}}^{(t)}) - f^\star \right] \leq \varepsilon \,, \text{ after } \tilde{\mathcal{O}} \left( \frac{\sigma^2}{n\varepsilon^2} + \frac{\sqrt{L}\sigma}{\sqrt{p}c\varepsilon^{3/2}} + \frac{L(1 + \tilde{R}_0^2 R_0^{-2})}{pc\varepsilon} \right) \cdot R_0^2 \text{ iterations,}$$

*where $\tilde{R}_0^2 = \frac{1}{n} \sum_{i=1}^{n} \|\mathbf{x}_i^{(0)} - \bar{\mathbf{x}}^{(0)}\|^2 + \frac{1}{nL^2} \sum_{i=1}^{n} \|\mathbf{y}_i^{(0)} - \bar{\mathbf{y}}^{(0)}\|^2$.*

From these results we see that the leading term in the convergence rate (assuming $\sigma > 0$) is not affected by the graph parameters. Moreover, in this term we see a linear speedup in $n$, the number of workers. The leading terms of all three results match with the convergence estimates for all-reduce mini-batch SGD [5, 43] and is optimal [37]. This means, that after a sufficiently long transient time, GT achieves a linear speedup in $n$. This transient time depends on the graph parameters $p$ and $c$, but not on the data-dissimilarity. We will discuss the dependency of the convergence rate on the graph parameters $c, p$ more carefully below in Sections 5 and 7, and compare the convergence rate to the convergence rates of D-SGD and $D^2$.

**Possible Relaxations of the Assumptions.** Before moving on to the proofs, we mention briefly a few possible relaxations of the assumptions that are possible with only slight adaptions of the proof framework. These extensions can be addressed with known techniques and are omitted for conciseness. We give here the necessary references for completeness.

- **Bounded Gradient Assumption I.** The uniform bound on the stochastic noise in Assumption 4 could be relaxed by allowing the noise to grow with the gradient norm [16, Assumption 3b].
- **Bounded Gradient Assumption II.** In the convex setting it has been observed that $\sigma^2$ can be replaced with $\sigma_\star^2 := \frac{1}{n} \sum_{i=1}^{n} \mathbb{E}_{\xi_i} \left\| \nabla F_i(\mathbf{x}^\star, \xi_i) - \nabla f_i(\mathbf{x}^\star) \right\|_2^2$, the noise at the optimum. However, this requires smoothness of each $F_i(\mathbf{x}, \xi)$, $\xi \in \mathcal{D}_i$, which is stronger than our Assumption 2. For the technique see e.g. [38].
- **Different mixing for $X$ and $Y$.** In Algorithm 1, both the $\mathbf{x}$ and $\mathbf{y}$ iterates are averaged on the same communication topology (the same mixing matrix). This can be relaxed by allowing for two separate matrices. This follows from inspecting our proof below.
- **Local Steps.** It is possible to extend Algorithm 1 and our analysis in Theorem 2 to allow for local computation steps. Mixing matrix would alternate between identity matrix $I$ (no communication, local steps) and $W$ (communication steps).
  However, it is non trivial to extend our analysis to the general time-varying graphs, as the product of two arbitrary mixing matrices $W_1 W_2$ might be non symmetric.

## 4.2 Faster convergence on consensus functions

We now state an additional result, which improves Theorem 2 on the consensus problem, defined as

$$\min \left[ f(\mathbf{x}) = \frac{1}{n} \sum_{i=1}^{n} \left[ f_i(\mathbf{x}) := \frac{1}{2} \|\mathbf{x} - \boldsymbol{\mu}_i\|^2 \right] \right], \tag{7}$$

for vectors $\boldsymbol{\mu}_i \in \mathbb{R}^d$, $i \in [n]$ and optimal solution $\mathbf{x}^\star = \frac{1}{n} \sum_{i=1}^{n} \boldsymbol{\mu}_i$. Note that this is a special case of the general problem (1) without stochastic noise ($\sigma = 0$). For this function, we can improve the complexity estimate that would follow from Theorem 2 by proving a convergence rate that does not depend on $c$.

**Theorem 3.** *Let $f$ be as in (7) let Assumption 1 hold. Then there exists a stepsize $\gamma \leq p$ such that it holds $\frac{1}{n} \sum_{i=1}^{n} \left\| \mathbf{x}_i^{(T)} - \mathbf{x}^\star \right\|^2 \leq \epsilon$, for the iterates GT 1 and any $\epsilon > 0$, after at most $T = \tilde{\mathcal{O}} \left( p \log \frac{1}{\epsilon} \right)$ iterations.*

# 5 Discussion

We now provide a discussion of these results.

## 5.1 Parameter $c$

The convergence rate in Theorem 2 depends on the parameter $c$, that in the worst case could be as small as $p$. In this case our theoretical result does not improve over existing results for the strongly convex case. However, for many graphs in practice parameter $c$ is bounded by a constant (see Table 3 and discussion below Assumption 1).

While we show in Theorem 3 that it is possible to remove the dependency on $c$ entirely from the convergence rate in special cases, it is still an open question if the parameter $c$ in Theorem 2 is tight in general.

## 5.2 Comparison to prior GT literature

Tables 1 and 2 compare our theoretical convergence rates in strongly convex and non convex settings. Our result tightens all existing prior work.

## 5.3 Comparison to other methods.

We now compare our complexity estimate of GT to D-SGD and $\mathrm{D}^2$ in the strongly convex case. Analogous observations hold for the other cases too.

**Comparison to D-SGD.** A popular algorithm for decentralized optimization is D-SGD [24] that converges as [16]:

$$\tilde{\mathcal{O}} \left( \frac{\sigma^2}{\mu n \varepsilon} + \frac{\sqrt{L} \left( \zeta + \sqrt{p} \sigma \right)}{\mu p \sqrt{\varepsilon}} + \frac{L}{\mu p} \log \frac{1}{\varepsilon} \right). \tag{D-SGD}$$

While GT is agnostic to data-heterogenity, here the convergence estimate depends on the data-heterogenity, measured by a constant $\zeta^2$ that satisfies:

$$\frac{1}{n} \sum_{i=1}^{n} \|\nabla f_i(\mathbf{x}^\star) - \nabla f(\mathbf{x}^\star)\|_2^2 \leq \zeta^2. \tag{8}$$

Comparing with Theorem 2, GT completely removes dependence on data heterogeneity level $\zeta$. Moreover, even in the homogeneous case when $\zeta = 0$, GT enjoys the same rate as D-SGD for many practical graphs when $c$ is bounded by a constant.

**Comparison to $\mathrm{D}^2$.** Similarly to GT, $\mathrm{D}^2$ also removes the dependence on functions heterogeneity. The convergence rate of $\mathrm{D}^2$ holds under assumption that $\lambda_{\min}(W) > -\frac{1}{3}$ and it is equal to [55]:

$$\mathcal{O} \left( \frac{\sigma^2}{\mu n \varepsilon} + \frac{\sqrt{L} \sigma}{\mu \sqrt{p} \sqrt{\varepsilon}} + \frac{L}{\mu p} \log \frac{1}{\varepsilon} \right). \tag{$\mathrm{D}^2$}$$

Under the assumption $\lambda_{\min}(W) > -\frac{1}{3}$ the parameter $c$ is a constant, and the GT rate estimated in Theorem 2 matches ($\mathrm{D}^2$).

# 6 Proof sketch of the main theorem

Here we give a proof sketch for Theorem 2, for the special case of strongly convex objectives. We give all proof details in the appendix and highlight the main technical difficulties and novel techniques.

**Key Lemma.** It is very common—and useful—to write the iterates in the form $X^{(t)} = \bar{X}^{(t)} + (X^{(t)} - \bar{X}^{(t)})$, where $\bar{X}^{(t)}$ denotes the matrix with the average over the nodes. We can then separately analyze $\bar{X}^{(t)}$ and the consensus difference $\Delta X^{(t)} := (X^{(t)} - \bar{X}^{(t)})$ (and $\Delta Y^{(t)} := (Y^{(t)} - \bar{Y}^{(t)})$). Define $\tilde{W} = W - \frac{\mathbf{1}\mathbf{1}^\top}{n}$. From the update equation (GT) we see that

$$\begin{pmatrix} \Delta X^{(t+1)} \\ \gamma \Delta Y^{(t+1)} \end{pmatrix}^\top = \underbrace{\begin{pmatrix} \Delta X^{(t)} \\ \gamma \Delta Y^{(t)} \end{pmatrix}^\top}_{=:\Psi_t} \underbrace{\begin{pmatrix} \tilde{W} & 0 \\ -\tilde{W} & \tilde{W} \end{pmatrix}}_{=:J} + \gamma \underbrace{\begin{pmatrix} 0 \\ \left(\nabla F(X^{t+1}, \xi^{t+1}) - \nabla F(X^t, \xi^t)\right)\left(I - \frac{\mathbf{1}\mathbf{1}^\top}{n}\right) \end{pmatrix}^\top}_{=:E_t},$$

in short, by using the notation $\Psi_t$, $J$, and $E_t$ as introduced above,

$$\Psi_{t+1} = \Psi_t J + \gamma E_t. \tag{9}$$

We could immediately adapt the proof technique from [16] if it would hold that the spectral radius of $J$ is smaller than one. However, this is not the case, and in general $\|J\| > 1$.

Note that for any integer $i \geq 0$:

$$J^i = \begin{pmatrix} \tilde{W}^i & 0 \\ -i\tilde{W}^i & \tilde{W}^i \end{pmatrix} \qquad \|J^i\|^2 = \|\tilde{W}^i\|^2 + i^2\|\tilde{W}^i\|^2 \leq (1-p)^i + i^2(1-p)^i, \tag{10}$$

by Assumption 1. With this observation we can now formulate a key lemma:

**Lemma 4** (Contraction). *For any integer $\tau \geq \frac{2}{p} \log\left(\frac{50}{p}(1 + \log\frac{1}{p})\right)$ it holds that $\|J^\tau\|^2 \leq \frac{1}{2}$.*

While the constants in this lemma are chosen to ease the presentation, most important for us is that after $\tau = \tilde{\Theta}\left(\frac{1}{p}\right)$ communication rounds, old parameter values (from $\tau$ steps ago) get discounted and averaged by a constant factor. We can alternatively write the statement of Lemma 4 as

$$\left\|ZJ^\tau - \bar{Z}\right\|_F^2 \leq \frac{1}{2}\left\|Z - \bar{Z}\right\|_F^2, \qquad \forall Z \in \mathbb{R}^{2d \times n}.$$

This resembles [16, Assumption 4] and the proof now follows the same pattern. A few crucial differences remain, as the result in [16] depends on a data-dissimilarity parameter which we can avoid by carefully estimating the tracking errors. For completeness, we sketch the outline and give all details in the appendix.

**Average Sequence.** First, we consider the average sequences $\bar{X}^{(t)}$ and $\bar{Y}^{(t)}$. As all columns of these matrices are equal, we can equivalently consider a single column only: $\bar{\mathbf{x}}^{(t)}$ and $\bar{\mathbf{y}}^{(t)}$.

**Lemma 5** (Average). *It holds that*

$$\bar{\mathbf{y}}^{(t)} = \frac{1}{n}\sum_{i=1}^n \nabla F_i\left(\mathbf{x}_i^{(t)}, \xi_i^{(t)}\right), \qquad \bar{\mathbf{x}}^{(t+1)} = \bar{\mathbf{x}}^{(t)} - \gamma\frac{1}{n}\sum_{i=1}^n \nabla F_i\left(\mathbf{x}_i^{(t)}, \xi_i^{(t)}\right). \tag{11}$$

This follows directly from the update (GT) and the fact that $\bar{X} = \bar{X}W$ for doubly stochastic mixing matrices. The update of $\bar{\mathbf{x}}^{(t)}$ in (11) is almost identical to one step of mini-batch SGD (on a complete graph). The average sequence behaves almost as a SGD sequence:

**Lemma 6** (Descent lemma, [16, Lemma 8]). *Under the Assumptions of Theorem 2 for the convex functions, the averages $\bar{\mathbf{x}}^{(t)} := \frac{1}{n}\sum_{i=1}^n \mathbf{x}_i^{(t)}$ of the iterates of Algorithm 1 with the stepsize $\gamma \leq \frac{1}{12L}$ satisfy*

$$\mathbb{E}\left\|\bar{\mathbf{x}}^{(t+1)} - \mathbf{x}^\star\right\|^2 \leq \left(1 - \frac{\gamma\mu}{2}\right)\mathbb{E}\left\|\bar{\mathbf{x}}^{(t)} - \mathbf{x}^\star\right\|^2 + \frac{\gamma^2\sigma^2}{n} - \gamma e_t + \frac{3\gamma L}{n}\sum_{i=1}^n \mathbb{E}\left\|\bar{\mathbf{x}}^{(t)} - \mathbf{x}_i^{(t)}\right\|^2, \tag{12}$$

*where $e_t = \mathbb{E}\,f(\bar{\mathbf{x}}^{(t)}) - f^\star$, for $f^\star = \min_{\mathbf{x} \in \mathbb{R}^d} f(\mathbf{x})$.*

**Consensus Distance.** The main difficulty comes from estimating the consensus distance $\|\Psi_t\|^2$, in the notation introduced in (9). Note that

$$\|\Psi_t\|^2 = \frac{1}{n}\sum_{i=1}^n \left\|\mathbf{x}_i^{(t)} - \bar{\mathbf{x}}^{(t)}\right\|_2^2 + \frac{\gamma^2}{n}\sum_{i=1}^n \left\|\mathbf{y}_i^{(t)} - \bar{\mathbf{y}}^{(t)}\right\|_2^2.$$

By unrolling (9) for $\tau \leq k \leq 2\tau$, $\tau = \frac{2}{p}\log\left(\frac{50}{p}(1 + \log\frac{1}{p})\right) + 1$ steps,

$$\Psi_{t+k} = \Psi_t J^k + \gamma\sum_{j=1}^{k-1} E_{t+j-1}J^{k-j}. \tag{13}$$

By taking the Frobenius norm, and carefully estimating the norm of the error term $\left\|\sum_{j=1}^{\tau-1} E_{t+j-1}J^{\tau-j}\right\|_F^2$, and using Lemma 4 we can derive a recursion for the consensus distance.

**Lemma 7** (Consensus distance recursion). *There exists absolute constants $B_1, B_2, B_3 > 0$ such that for a stepsize $\gamma < \frac{c}{B_3 L\tau}$*

$$\mathbb{E}\|\Psi_{t+k}\|_F^2 \leq \frac{7}{8}\mathbb{E}\|\Psi_t\|_F^2 + \frac{1}{128\tau}\sum_{j=0}^{k-1}\|\Psi_{t+j}\|_F^2 + \frac{B_1\tau L\gamma^2}{c^2}\sum_{j=0}^{k-1} ne_{t+j} + \frac{B_2\tau\gamma^2}{c^2}n\sigma^2. \tag{14}$$

This lemma allows to replace $p$ with $c$ in the final convergence rate. This is achieved by grouping same gradients in the sum $\left\|\sum_{j=1}^{k-1} E_{t+j-1}J^{k-j}\right\|_F^2$ and estimating the norm with Lemma 13.

An additional technical difficulty comes when unrolling consensus recursion (14). As iteration matrix $J$ is not contractive, i.e. $\|J\| > 1$, then $\|\Psi_{t+j}\|_F^2$ for $j < \tau$ can be larger than $\|\Psi_t\|_F^2$ (up to $\approx \frac{1}{p^2}$ times as $\|J^i\|^2 \leq \mathcal{O}\left(\frac{1}{p^2}\right)$ $\forall i$). We introduce an additional term in the recursion that is provably non-increasing

$$\Phi_{t+\tau} := \frac{1}{\tau}\sum_{j=0}^{\tau-1}\|\Psi_{t+j}\|_F^2.$$

With this we unroll consensus recursion.

**Lemma 8** (Unrolling recursion). *For $\gamma < \frac{c}{\sqrt{7B_1}L\tau} \leq \frac{1}{2L\tau}$ it holds,*

$$\mathbb{E}\|\Psi_t\|_F^2 \leq \left(1 - \frac{1}{64\tau}\right)^t A_0 + \frac{22B_1\tau L\gamma^2}{c^2}\sum_{j=0}^{t-1}\left(1 - \frac{1}{64\tau}\right)^{t-j} ne_j + \frac{20B_2\tau\gamma^2}{c^2}n\sigma^2 \tag{15}$$

*where $e_j = \mathbb{E}[f(\bar{\mathbf{x}}^{(j)}) - f(\mathbf{x}^\star)]$, $A_0 = 16\|\Delta X^{(0)}\|_F^2 + \frac{24\gamma^2}{p^2}\|\Delta Y^{(0)}\|_F^2$.*

It remains to combine (14) and (15) using technique from [16]. $\qquad\square$

**Proof sketch of Theorem 3.** Using the matrix notation introduced above, the iterations of GT on problem (7) can be written in a simple form:

$$\begin{pmatrix} \Delta X^{(t+1)} \\ \gamma\Delta Y^{(t+1)} \end{pmatrix}^\top = \begin{pmatrix} \Delta X^{(t)} \\ \gamma\Delta Y^{(t)} \end{pmatrix}^\top \underbrace{\begin{pmatrix} \tilde{W} & \gamma(W - I) \\ -\tilde{W} & (1-\gamma)\tilde{W} \end{pmatrix}}_{J'}.$$

Similar as above, also the matrix $J'$ is not a contraction operator, but in contrast to $J$ it is diagonalizable: $J' = Q\Lambda Q^{-1}$ for some $Q$ and diagonal $\Lambda$. It follows that $\|(J')^t\|^2 = \|Q\Lambda^t Q^{-1}\|^2$ is decreasing as $(1-p)^t\|Q\|^2\|Q^{-1}\|^2$. With this observation, the proof simplifies. $\qquad\square$

## 7 Experiments

In this section we investigate the tightness of parameters $c$ and $p$ in our theoretical result.

**Setup.** We consider simple quadratic functions defined as $f_i(\mathbf{x}) = \|\mathbf{x}\|^2$, and $\mathbf{x}^{(0)}$ is randomly initialized from a normal distribution $\mathcal{N}(0, 1)$. We add artificially stochastic noise to gradients as

$\nabla F_i(\mathbf{x}, \xi) = \nabla f_i(\mathbf{x}) + \xi$, where $\xi \sim \mathcal{N}(0, \frac{\sigma^2}{d}I)$ so that Assumption 4 is satisfied. We elaborate the details as well as results under other problem setups in Appendix C.

We verify the dependence on graph parameters $p$ and $c$ for the stochastic noise term. We fix the stepsize $\gamma$ to be constant, vary $p$ and $c$ and measure the value of $f(\bar{\mathbf{x}}^{(t)}) - f^\star$ that GT reaches after a large number of steps. According to the theory, GT converges to the level $\mathcal{O}\left(\frac{\gamma\sigma^2}{n} + \frac{\gamma^2\sigma^2}{pc^2}\right)$ in a linear number of steps (to reach higher accuracy, smaller stepsizes must be used). To decouple the second term we need to ensure that the first term is small enough. For that, we take the number of nodes $n$ to be large. In all experiments we ensure that the first term is at least by order of magnitude smaller than the second by comparing the noise level with GT on a fully-connected topology.

**The effect of $p$.** First, in Figure 1 we verify the expected $\mathcal{O}\left(\frac{1}{p}\right)$ dependence when $c$ is a constant. For a fixed $n = 300$ number of nodes with $d = 100$ we vary the value of a parameter $p$ by interpolating the ring topology (with uniform weights) with the fully-connected graph. The loss value $f(\mathbf{x}^{(\infty)})$ scales linearly in $\frac{1}{p}$ as can be observed in Figure 1 and the dependency on $p$ can thus not further be improved.

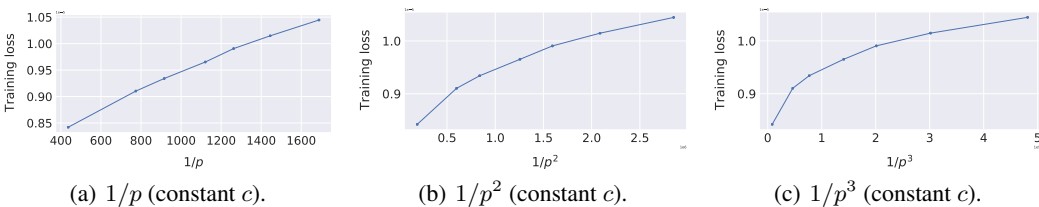

(a) $1/p$ (constant $c$).     (b) $1/p^2$ (constant $c$).     (c) $1/p^3$ (constant $c$).

Figure 1: Impact of $p$ on convergence with the stochastic noise $\sigma^2 = 1$, when $c$ and $\gamma$ are kept constant. We see a linear scaling in $\frac{1}{p}$ that verifies the $\mathcal{O}\left(\frac{1}{p}\right)$, dependence rather than prior predicted $\mathcal{O}\left(\frac{1}{p^2}\right)$.

**The effect of $c$.** In Figure 2 we aim to examine the dependence of the term $\mathcal{O}\left(\frac{1}{pc^2}\right)$ on the parameter $c$, in terms of $1/(pc^2)$ and $1/(cp)$. We take the ring topology on a fixed number of $n = 300$ nodes and reduce the self-weights to achieve different values of $c$ (see appendix for details). Otherwise the setup is as above. The current numerical results may suggest the existence of a potentially better theoretical dependence of the term $c$ (as discussed in Section 4.2); we leave the study for future work.

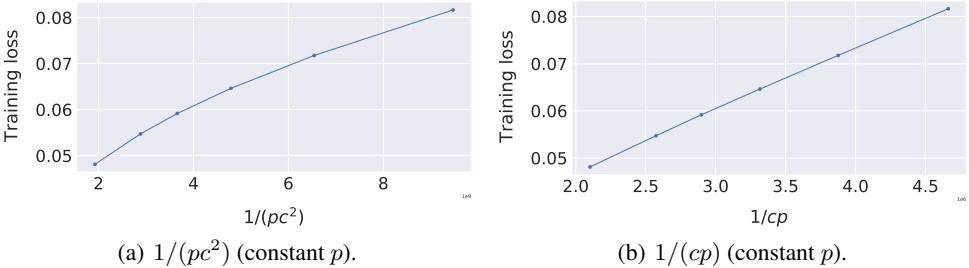

(a) $1/(pc^2)$ (constant $p$).       (b) $1/(cp)$ (constant $p$).

Figure 2: Impact of $c$ on the convergence with the stochastic noise $\sigma^2 = 1$, when $p$ and $\gamma$ are kept constant. We see a near linear scaling in $\mathcal{O}\left(\frac{1}{pc}\right)$ while the estimate $\mathcal{O}\left(\frac{1}{pc^2}\right)$ appears to be too conservative on this problem.

## 8 Conclusion

We have derived improved complexity bounds for the GT method, that improve over all previous results. We verify the tightness of the second term in the convergence rate in numerical experiments. Our analysis identifies that the smallest eigenvalue of the mixing matrix has a strong impact on the performance of GT, however the smallest eigenvalue can often be controlled in practice by choosing large enough self-weights ($w_{ii}$) on the nodes.

Our proof technique might be of independent interest in the community and might lead to improved analyses for other gossip based methods where the mixing matrix is not contracting (for e.g. in directed graphs, or using row- or column-stochastic matrices).

## Acknowledgments and Disclosure of Funding

This project was supported by SNSF grant 200020_200342, EU project DIGIPREDICT, and a Google PhD Fellowship. The authors thank Martin Jaggi for his support.

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
