\|^2$, where $\mathbf{x}, \boldsymbol{\mu}_i \in \mathbb{R}^d$. Then $\nabla f_i(\mathbf{x}) = \mathbf{x} - \boldsymbol{\mu}_i$. In matrix notation, the GT algorithm in this special case is equivalent to

$$\begin{pmatrix} X^{(t+1)} \\ \gamma Y^{(t+1)} \end{pmatrix}^{\top} = \begin{pmatrix} X^{(t)} \\ \gamma Y^{(t)} \end{pmatrix}^{\top} \begin{pmatrix} W & -W \\ 0 & W \end{pmatrix} + \gamma \begin{pmatrix} 0 \\ X^{(t+1)} - X^{(t)} \end{pmatrix}^{\top} = \begin{pmatrix} X^{(t)} \\ \gamma Y^{(t)} \end{pmatrix}^{\top} \begin{pmatrix} W & -W \\ \gamma(W - I) & (1 - \gamma)W \end{pmatrix}.$$

The optimal point $\mathbf{x}^{\star} = \bar{\boldsymbol{\mu}} = \frac{1}{n} \sum_{i=1}^{n} \boldsymbol{\mu}_i$. Denote $X^{\star} = [\mathbf{x}^{\star}, \ldots, \mathbf{x}^{\star}] \in \mathbb{R}^{d \times n}$. We decompose the error as

$$\left\| X^{(t)} - X^{\star} \right\|_F^2 = \underbrace{\left\| X^{(t)} - \bar{X}^{(t)} \right\|_F^2}_{\text{consensus error}} + \underbrace{\left\| \bar{X}^{(t)} - X^{\star} \right\|^2}_{\text{optimization error}}.$$

**For the optimization part,** notice that $\bar{Y}^{(t)} = \bar{X}^{(t)} - X^{\star}$. That is because

$$\bar{Y}^{(0)} = \nabla f(X^{(0)}) \frac{1}{n} \mathbf{1}\mathbf{1}^{\top} = \bar{X}^{(0)} - X^{\star}, \qquad \bar{Y}^{(t+1)} = \bar{Y}^{(t)} + \bar{X}^{(t+1)} - \bar{X}^{(t)}.$$

Therefore, the optimization error is equal to

$$\left\| \bar{X}^t - X^{\star} \right\|_F^2 = \left\| \bar{X}^{(t-1)} - \gamma \bar{Y}^{(t-1)} - X^{\star} \right\|_F^2 = \left\| (1 - \gamma) \left( \bar{X}^{(t-1)} - X^{\star} \right) \right\|_F^2$$
$$= (1 - \gamma)^{2t} \left\| \bar{X}^{(0)} - X^{\star} \right\|_F^2.$$

**For the consensus part,** denoting, $\tilde{W} = W - \frac{\mathbf{1}\mathbf{1}^{\top}}{n}, \Delta X^{(t)} = X^{(t)} - \bar{X}^{(t)}, \Delta Y^{(t)} = Y^{(t)} - \bar{Y}^{(t)}$,

$$\begin{pmatrix} \Delta X^{(t)} \\ \gamma \Delta Y^{(t)} \end{pmatrix}^{\top} = \begin{pmatrix} \Delta X^{(0)} \\ \gamma \Delta Y^{(0)} \end{pmatrix}^{\top} \underbrace{\begin{pmatrix} \tilde{W} & -\tilde{W} \\ \gamma(W - I) & (1 - \gamma)\tilde{W} \end{pmatrix}^{t}}_{J'}.$$

Taking the norm,

$$\left\| \Delta X^{(t)} \right\|_F^2 + \gamma^2 \left\| \Delta Y^{(t)} \right\|_F^2 \leq \|J'^t\|_2^2 \left( \left\| \Delta X^{(0)} \right\|_F^2 + \gamma^2 \left\| \Delta Y^{(0)} \right\|_F^2 \right).$$

Lets analyze spectral properties of matrix $J'^t$. Let the eigenvalue decomposition of $W$ be $W = U \Lambda U^{\top}$, the eigenvalue decomposition of $\tilde{W}$ is $\tilde{W} = U \tilde{\Lambda} U^{\top}$ for diagonal $\tilde{\Lambda}$.

We can decompose

$$J' = \begin{pmatrix} U & 0 \\ 0 & U \end{pmatrix} \underbrace{\begin{pmatrix} \tilde{\Lambda} & -\tilde{\Lambda} \\ \gamma(\Lambda - I) & (1 - \gamma)\tilde{\Lambda} \end{pmatrix}}_{=:M} \begin{pmatrix} U^{\top} & 0 \\ 0 & U^{\top} \end{pmatrix}.$$

And,

$$\|J'^t\|_2^2 = \left\| \begin{pmatrix} U & 0 \\ 0 & U \end{pmatrix} \begin{pmatrix} \tilde{\Lambda} & -\tilde{\Lambda} \\ \gamma(\Lambda - I) & (1 - \gamma)\tilde{\Lambda} \end{pmatrix}^{t} \begin{pmatrix} U^{\top} & 0 \\ 0 & U^{\top} \end{pmatrix} \right\|_2^2 = \left\| \begin{pmatrix} \tilde{\Lambda} & -\tilde{\Lambda} \\ \gamma(\Lambda - I) & (1 - \gamma)\tilde{\Lambda} \end{pmatrix}^{t} \right\|_2^2,$$

where the last equality is due to unitary property of $U$.

**Lemma 9.** *To diagonalize a block-diagonal matrix*

$$\begin{pmatrix} A & B \\ C & D \end{pmatrix},$$

where $A = \mathrm{diag}(a_0, \ldots a_n) \in R^{n \times n}$, $B = \mathrm{diag}(b_0, \ldots, b_n)$, $C = \mathrm{diag}(c_0, \ldots, c_n)$, $D = \mathrm{diag}(d_0, \ldots, d_n)$. *Assume that each of the $2 \times 2$ matrices*

$$\begin{pmatrix} a_i & b_i \\ c_i & d_i \end{pmatrix}$$

*are diagonalizable with*

$$\begin{pmatrix} a_i & b_i \\ c_i & d_i \end{pmatrix} = \begin{pmatrix} q_i^{(1)} & q_i^{(2)} \\ q_i^{(3)} & q_i^{(4)} \end{pmatrix} \cdot \begin{pmatrix} d_i^{(1)} & 0 \\ 0 & d_i^{(2)} \end{pmatrix} \cdot \begin{pmatrix} q_i^{(-1)} & q_i^{(-2)} \\ q_i^{(-3)} & q_i^{(-4)} \end{pmatrix}$$

*Then the original matrix is diagonalizable and its diagonalization is equal to*

$$\begin{pmatrix} A & B \\ C & D \end{pmatrix} = \begin{pmatrix} Q_1 & Q_2 \\ Q_3 & Q_4 \end{pmatrix} \cdot \begin{pmatrix} D_1 & 0 \\ 0 & D_2 \end{pmatrix} \cdot \begin{pmatrix} Q_{-1} & Q_{-2} \\ Q_{-3} & Q_{-4} \end{pmatrix},$$

*where each $Q_l = \mathrm{diag}\left(q_1^{(l)}, \ldots, q_n^{(l)}\right)$, $D_l = \mathrm{diag}\left(d_1^{(l)}, \ldots d_n^{(l)}\right)$.*

We need to show that the following $2 \times 2$ matrices are diagonalizable.

$$M_i := \begin{pmatrix} \lambda_i & -\lambda_i \\ \gamma(\lambda_i - 1) & (1 - \gamma)\lambda_i \end{pmatrix},$$

where the $\lambda_i$ are eigenvalues of the matrix $\tilde{W}$. The eigenvalues of $M_i$ are

$$\lambda(M_i) = \left\{ \lambda_i - \frac{\gamma\lambda_i}{2} - \frac{1}{2}\sqrt{\gamma\lambda_i}\sqrt{4 + (\gamma - 4)\lambda_i}, \lambda_i - \frac{\gamma\lambda_i}{2} + \frac{1}{2}\sqrt{\gamma\lambda_i}\sqrt{4 + (\gamma - 4)\lambda_i} \right\},$$

which are distinct for $\gamma > 0$, therefore the matrix is diagonalizble (over $\mathbb{C}$).

If $\lambda_i$ is positive, then by choosing $\gamma \leq 1 - \lambda_i$,

$$|\lambda(M_i)| \leq \frac{1}{3}\lambda_i + \frac{2}{3}.$$

If $\lambda_i$ is negative, then, then by choosing $\gamma \leq 1 - |\lambda_i|$,

$$|\lambda(M_i)| \leq \frac{1}{3}|\lambda_i| + \frac{2}{3}.$$

We do not give the full formal prove of these two bounds. First we note that $|(M_i)|$ is monotone in $\gamma$, i.e. the absolute value increases in $\gamma$. Therefore it is enough to check that it holds $|\lambda(M_i)| \leq \frac{1}{3}|\lambda_i| + \frac{2}{3}$ for $\gamma = 1 - |\lambda_i|$. We visualize these upper bounds with Mathematica [9] in Figure 3.

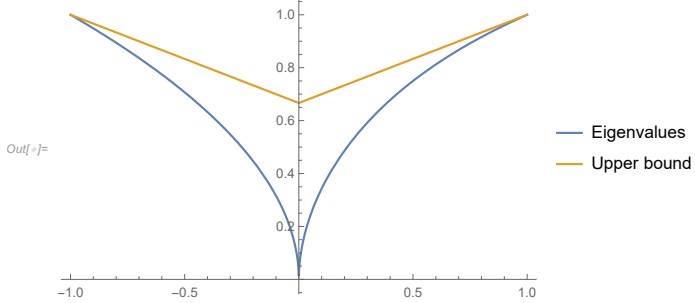

Figure 3: The upper bound $\frac{1}{3}|\lambda_i| + \frac{2}{3}$ (yellow) vs. the true $|\lambda(M_i)|$ for the choice $\gamma = 1 - |\lambda_i|$.

This concludes the proof.

# B  Proof of Theorem 2 — General Case

We first re-state theorem 2 in terms of number of iterations $T$

**Theorem 10.** *For GT algorithm 1 with a mixing matrix as in Definition 1, under Assumptions 1, 2, 4, after $T$ iterations, if $T > \frac{2}{p} \log \left( \frac{50}{p} (1 + \log \frac{1}{p}) \right)$, there exists a constant stepsize $\gamma_t = \gamma$ such that the error is bounded as*
**Non-convex:**

$$\frac{1}{T+1} \sum_{t=0}^{T} \left\| \nabla f(\bar{\mathbf{x}}^{(t)}) \right\|_2^2 \leq \tilde{\mathcal{O}} \left( \sqrt{\frac{LF_0\sigma^2}{nT}} + \left( \frac{\sigma LF_0}{(\sqrt{p}c + p\sqrt{n})T} \right)^{2/3} + \frac{L(F_0 + L\tilde{R}_0^2)}{pcT} \right),$$

**Strongly-convex:** *Under additional Assumption 3 with $\mu > 0$, it holds*

$$\sum_{t=0}^{T} \frac{w_t}{W_T} \left[ \mathbb{E} f(\bar{\mathbf{x}}^{(t)}) - f^\star \right] + \frac{\mu}{2} R_{T+1} \leq \tilde{\mathcal{O}} \left( \frac{\sigma^2}{\mu nT} + \frac{L\sigma^2}{\mu^2 pc^2 T^2} + \frac{L(R_0^2 + \frac{L}{\mu}\tilde{R}_0^2)}{pc} \exp \left[ -\frac{\mu pcT}{L} \right] \right),$$

**Weakly-convex:** *Under Assumptions 3 with $\mu \geq 0$, it holds*

$$\frac{1}{T+1} \sum_{t=0}^{T} \left[ \mathbb{E} f(\bar{\mathbf{x}}^{(t)}) - f^\star \right] \leq \tilde{\mathcal{O}} \left( \sqrt{\frac{R_0^2\sigma^2}{nT}} + \left( \frac{\sigma\sqrt{L}R_0^2}{\sqrt{p}cT} \right)^{2/3} + \frac{L(R_0^2 + \tilde{R}_0^2)}{pcT} \right),$$

*where $F_0 = f(\bar{\mathbf{x}}^{(0)}) - f^\star$, $R_t = \left\| \mathbf{x}^{(t)} - \mathbf{x}^\star \right\|$, $t \in \{0, T+1\}$, $\tilde{R}_0^2 = \frac{1}{n} \sum_{i=1}^{n} \left\| \mathbf{x}_i^{(0)} - \bar{\mathbf{x}}^{(0)} \right\|^2 + \frac{1}{n} \sum_{i=1}^{n} \left\| \mathbf{y}_i^{(0)} - \bar{\mathbf{y}}^{(0)} \right\|^2$.*

## B.1  Useful Inequalities

*Proof of Lemma 4.* By monotonicity, it suffices to check the inequality for $i = \tau$. By using $(1-p)^i \leq e^{-ip}$ and plugging $\tau$ into (10) it follows:

$$\left\| J^i \right\|^2 \leq e^{-\tau p} (1 + \tau^2) \leq \frac{p^2}{50^2 (1 + \log \frac{1}{p})^2} \left( 1 + \frac{(2(\log(50) + \log(\frac{1}{p}(1 + \log \frac{1}{p})))^2}{p^2} \right)$$

$$\leq \frac{1}{50^2} + \frac{1}{10} + \frac{1}{4}$$

with $\log(\frac{1}{p}(1 + \log \frac{1}{p}) \leq \log \frac{1}{p} + \log \log \frac{1}{p} \leq 2\log \frac{1}{p}$, then $(\log(4) + 2\log \frac{1}{p})^2 \leq 2\log 4 + 8\log \frac{1}{p}$, and $(4\log 50 + 16\log \frac{1}{p}))^2 \leq (128 + 512\log \frac{1}{p})$. $\square$

**Lemma 11.** *Let $\lambda \in (-1, 1)$ with $|\lambda| = 1 - \alpha$, for $0 < \alpha < 1$. Then $\left| i\lambda^i \right| \leq \frac{1}{\alpha}$ for all $i \geq 0$.*

*Proof.*
$$\left| i\lambda^i \right| \leq i(1-\alpha)^i \leq \sum_{j=1}^{i} (1-\alpha)^j \leq \frac{1-\alpha}{\alpha}. \qquad \square$$

**Lemma 12** (fact). *Let $W$ be a symmetric matrix with eigenvalues $\lambda_1(W) \geq \ldots \lambda_n(W)$. Then $\|W\|^2 = \max_i \lambda_i^2(W)$.*

**Lemma 13.** *It holds $\left\| (i+1)\tilde{W}^{i+1} - i\tilde{W}^i \right\|^2 \leq \frac{4}{\alpha^2} \leq \frac{16}{c^2}$ for all $i \geq 0$, where $\alpha = 1 - |\lambda_n(W)|$ and $c$ as defined in (3).*

*Proof.* The eigenvalues of $(i+1)\tilde{W}^{i+1} - i\tilde{W}^i$ have the form $(i+1)\lambda^{i+1} - i\lambda^i$, for $\lambda \in \Lambda := \{\lambda_1(\tilde{W}), \ldots, \lambda_n(\tilde{W})\}$, the eigenvalues of $\tilde{W}$. By Lemma 12, it holds

$$\left\| (i+1)\tilde{W}^{i+1} - i\tilde{W}^i \right\|^2 = \max_{\lambda \in \Lambda} ((i+1)\lambda^{i+1} - i\lambda^i)^2.$$

If the maximum is attained for a positive $\lambda > 0$, we conclude
$$((i+1)\lambda^{i+1} - i\lambda^i)^2 = (\lambda^{i+1} - i\lambda^i(1-\lambda))^2$$
$$\leq 2(\lambda^{i+1})^2 + 2(1-\lambda)^2(i\lambda^i)^2$$
$$\leq 2(\lambda^{i+1})^2 + 2\frac{(1-\lambda)^2}{(1-\lambda)^2}$$
$$\leq 4$$

with Lemma 11 for the first estimate and using $\lambda \leq 1$ on the last line. If the maximum is attained for a negative $\lambda < 0$ with $\lambda = -1 + \beta$, for $\beta > 0$, then
$$((i+1)\lambda^{i+1} - i\lambda^i)^2 \leq 2((i+1)\lambda^{i+1})^2 + 2(i\lambda^i)^2$$
$$\leq \frac{2}{\beta^2} + \frac{2}{\beta^2} \leq \frac{4}{\alpha^2}$$

with Lemma 11 and $\alpha \leq \beta$.

Note that $c = 1 - (1-\alpha)^2 = 2\alpha - \alpha^2 \geq \alpha$, since $\alpha(1-\alpha) \geq 0$ and that $c \leq 2\alpha$. $\qquad\square$

**Lemma 14.** *It holds* $\left\| i\tilde{W}^i \right\|^2 \leq \frac{1}{\alpha^2} \leq \frac{4}{p^2}$.

*Proof.* $\left\| i\tilde{W}^i \right\|^2 = \left( i \left\| \tilde{W}^i \right\| \right)^2$, and the proof follows with Lemma 11 and 12 from above. $\qquad\square$

**Lemma 15.** *It holds* $\left\| \Psi^0 J^t \right\|_F^2 \leq 2 \left\| \Delta X^{(0)} \right\|_F^2 + \frac{3\gamma^2}{p^2} \left\| \Delta Y^{(0)} \right\|_F^2$ *for all* $t \geq 0$, *where $p$ is defined in* (3).

*Proof.* Starting from (10) and using Lemma 11 with $\delta = 1 - \lambda_2$
$$\left\| \Psi^0 J^i \right\|_F^2 = \left\| \begin{pmatrix} \Delta X^{(0)}\tilde{W}^i - i\gamma\Delta Y^{(0)}\tilde{W}^i \\ \gamma\Delta Y^{(0)}\tilde{W}^i \end{pmatrix}^\top \right\|_F^2 \leq 2 \left\| \Delta X^{(0)} \right\|_F^2 + \frac{3\gamma^2}{p^2} \left\| \Delta Y^{(0)} \right\|_F^2.$$
$\qquad\square$

**Lemma 16.** *For arbitrary set of $n$ vectors* $\{\mathbf{a}_i\}_{i=1}^n$, $\mathbf{a}_i \in \mathbb{R}^d$
$$\left\| \sum_{i=1}^n \mathbf{a}_i \right\|^2 \leq n \sum_{i=1}^n \|\mathbf{a}_i\|^2. \tag{16}$$

**Lemma 17.** *For given two vectors* $\mathbf{a}, \mathbf{b} \in \mathbb{R}^d$
$$2\langle \mathbf{a}, \mathbf{b} \rangle \leq \gamma \|\mathbf{a}\|^2 + \gamma^{-1}\|\mathbf{b}\|^2, \qquad\qquad \forall\gamma > 0. \tag{17}$$

**Lemma 18.** *For given two vectors* $\mathbf{a}, \mathbf{b} \in \mathbb{R}^d$
$$\|\mathbf{a} + \mathbf{b}\|^2 \leq (1+\alpha)\|\mathbf{a}\|^2 + (1 + \alpha^{-1})\|\mathbf{b}\|^2, \qquad\qquad \forall\alpha > 0. \tag{18}$$
*This inequality also holds for the sum of two matrices $A, B \in \mathbb{R}^{n \times d}$ in Frobenius norm.*

**Lemma 19.** *For* $A \in \mathbb{R}^{d \times n}$, $B \in \mathbb{R}^{n \times n}$
$$\|AB\|_F \leq \|A\|_F \|B\|_2. \tag{19}$$

## B.2 Convex Cases

**Proof of Lemma 7** We first state auxiliary lemma about consensus recursion.

**Lemma 20.** *There exists absolute constants $C_1 = 440, C_2 = 380$ such that iterates of Algorithm 1 satisfy,*
$$\mathbb{E}\|\Psi_{t+k}\|_F^2 \leq \frac{3}{4}\mathbb{E}\|\Psi_t\|_F^2 + \gamma^2 \frac{C_1\tau}{c^2}\sum_{j=0}^{k-1}\mathbb{E}\left\|\nabla f(X^{t+j}) - \nabla f(X^\star)\right\|_F^2 + \gamma^2 \frac{C_2\tau}{c^2}n\sigma^2. \tag{20}$$

*where* $\tau \leq k \leq 2\tau$, $\tau = \frac{2}{p}\log\left(\frac{50}{p}(1 + \log\frac{1}{p})\right) + 1$, *$p$ and $c$ are defined in* (3), $\Psi_t = \left(\Delta X^{(t)}, \gamma\Delta Y^{(t)}\right)$ *and is defined in* (9).

*Proof.* We start from the recursion (13) given in the main text

$$\Psi_{t+k} = \Psi_t J^k + \gamma \sum_{j=1}^{k} E_{t+j-1} J^{k-j} \,.$$

Taking the norm,

$$\|\Psi_{t+k}\|_F^2 \overset{(18),\alpha=\frac{1}{4},(19)}{\leq} \left(1 + \frac{1}{4}\right) \|J^k\|_2^2 \|\Psi_t\|_F^2 + 5\gamma^2 \left\| \sum_{j=1}^{k} E_{t+j-1} J^{k-j} \right\|_F^2$$

Using the key Lemma 4, the first term can be estimated as

$$\left(1 + \frac{1}{4}\right) \|J^k\|_2^2 \|\Psi_t\|_F^2 \leq \frac{3}{4} \|\Psi_t\|_F^2 \,.$$

Lets estimate separately the second term. Denoting $G^{(t)} = \nabla F(X^{(t)}, \xi^{(t)})$,

$$\left\| \sum_{j=1}^{k} E_{t+j-1} J^{k-j} \right\|_F^2 = \left\| \begin{pmatrix} -\sum_{j=1}^{k} \left(G^{(t+j)} - G^{(t+j-1)}\right)(k-j)\tilde{W}^{k-j}(I - \frac{\mathbf{1}\mathbf{1}^\top}{n}) \\ \sum_{j=1}^{k} \left(G^{(t+j)} - G^{(t+j-1)}\right)\tilde{W}^{k-j}(I - \frac{\mathbf{1}\mathbf{1}^\top}{n}) \end{pmatrix} \right\|_F^2$$

$$\overset{(19)}{\leq} \underbrace{\left\| \sum_{j=1}^{k} \left(G^{(t+j)} - G^{(t+j-1)}\right)(k-j)\tilde{W}^{k-j} \right\|_F^2}_{=:T_1}$$

$$+ \underbrace{\left\| \sum_{j=1}^{k} \left(G^{(t+j)} - G^{(t+j-1)}\right)\tilde{W}^{k-j} \right\|_F^2}_{=:T_2},$$

where we used the definition of the Frobenius norm and $\left\| I - \frac{\mathbf{1}\mathbf{1}^\top}{n} \right\| \leq 1$. We now give upper bounds for $T_1$ and $T_2$.

**The second term $T_2$.** We firstly separate the stochastic noise by adding and subtracting the full gradient,

$$T_2 \overset{(18)}{\leq} 3 \left\| \sum_{j=1}^{k} \left(\nabla f(X^{(t+j)}) - \nabla f(X^{(t+j-1)})\right)\tilde{W}^{k-j} \right\|_F^2$$

$$+ 3 \left\| \sum_{j=1}^{k} \left(G^{(t+j)} - \nabla f(X^{(t+j)})\right)\tilde{W}^{k-j} \right\|_F^2 + 3 \left\| \sum_{j=1}^{k} \left(G^{(t+j-1)} - \nabla f(X^{(t+j-1)})\right)\tilde{W}^{k-j} \right\|_F^2.$$

Note that

$$\mathbb{E} \left\| \sum_{j=1}^{k} \left(G^{(t+j)} - \nabla f(X^{(t+j)})\right)\tilde{W}^{k-j} \right\|_F^2 = \sum_{j=1}^{k} \mathbb{E} \left\| \left(G^{(t+j)} - \nabla f(X^{(t+j)})\right)\tilde{W}^{k-j} \right\|_F^2$$

$$\overset{\|\tilde{W}\|\leq 1}{\leq} \sum_{j=1}^{k} \mathbb{E} \left\| G^{(t+j)} - \nabla f(X^{(t+j)}) \right\|_F^2 \,,$$

where we used the martingale property $\mathbb{E}_j \left[ G^{(j)} - \nabla f(X^{(j)}) \mid X^{(j)} \right] = 0$ for all $j \leq t$. It follows

$$\mathbb{E}[T_2] \overset{(6)}{\leq} 3\,\mathbb{E} \left\| \sum_{j=1}^{k} \left(\nabla f(X^{(t+j)}) - \nabla f(X^{(t+j-1)})\right)\tilde{W}^{k-j} \right\|_F^2 + 6kn\sigma^2 \,.$$

We expand further by adding and subtracting $\nabla f(X^\star)$ to the first norm, and bounding stochastic noise by (6) in the other terms

$$
\mathbb{E}[T_2] \overset{(18),(6)}{\leq} 6\,\mathbb{E}\left\|\sum_{j=1}^{k}(\nabla f(X^{(t+j)}) - \nabla f(X^\star))\tilde{W}^{k-j}\right\|_F^2 + 6\,\mathbb{E}\left\|\sum_{j=1}^{k}(\nabla f(X^{(t+j-1)}) - \nabla f(X^\star))\tilde{W}^{k-j}\right\|_F^2 + 6kn\sigma^2
$$

$$
\overset{(16),(19)}{\leq} 12k\sum_{j=0}^{k}\mathbb{E}\left\|\nabla f(X^{(t+j)}) - \nabla f(X^\star)\right\|_F^2 + 6kn\sigma^2\,.
$$

**The first term $T_1$.** First, we separate the stochastic noise similarly as above. Defining $Z^{(t)} = G^{(t)} - \nabla f(X^{(t)})$,

$$
T_1 \overset{(18)}{\leq} 2\left\|\sum_{j=1}^{k}\left[\nabla f(X^{(t+j)}) - \nabla f(X^{(t+j-1)})\right](k-j)\tilde{W}^{k-j}\right\|_F^2 + 2\left\|\sum_{j=1}^{k}\left(Z^{(t+j)} - Z^{(t+j-1)}\right)(k-j)\tilde{W}^{k-j}\right\|_F^2\,.
$$

Next, we add and subtract $\nabla f(X^\star)$ in the first term $k-1$ times and temporarily denote $D^{(j)} = \nabla f(X^{(j)}) - \nabla f(X^\star)$

$$
T_1 \leq 2\left\|\sum_{j=1}^{k}\left(D^{(t+j)} - D^{(t+j-1)}\right)(k-j)\tilde{W}^{k-j}\right\|_F^2 + 2\left\|\sum_{j=1}^{k}\left(Z^{(t+j)} - Z^{(t+j-1)}\right)(k-j)\tilde{W}^{k-j}\right\|_F^2\,.
$$

Next, we re-group the sums by the gradient index.

$$
T_1 \leq 2\left\|D^{(t+k-1)}\tilde{W} - (k-1)D^{(t)}\tilde{W}^{k-1} + \sum_{j=1}^{k-2}D^{(t+j)}\left[(k-j)\tilde{W}^{k-j} - (k-j-1)\tilde{W}^{k-j-1}\right]\right\|_F^2
$$

$$
+ 2\left\|Z^{(t+k-1)}\tilde{W} - (k-1)Z^{(t)}\tilde{W}^{k-1} + \sum_{j=1}^{k-2}Z^{(t+j)}\left[(k-j)\tilde{W}^{k-j} - (k-j-1)\tilde{W}^{k-j-1}\right]\right\|_F^2
$$

$$
\overset{(16),(19)}{\leq} 2k\left[\left\|D^{(t+k-1)}\right\|_F^2 + \left\|D^{(t)}(k-1)\tilde{W}^{k-1}\right\|_F^2 + \sum_{j=1}^{k-2}\left\|D^{(t+j)}\left[(k-j)\tilde{W}^{k-j} - (k-j-1)\tilde{W}^{k-j-1}\right]\right\|_F^2\right]
$$

$$
+ 2\left[\left\|Z^{(t+k-1)}\right\|_F^2 + \left\|Z^{(t)}(k-1)\tilde{W}^{k-1}\right\|_F^2 + \sum_{j=1}^{k-2}\left\|Z^{(t+j)}\left[(k-j)\tilde{W}^{k-j} - (k-j-1)\tilde{W}^{k-j-1}\right]\right\|_F^2\right]
$$

where for splitting $Z$ we used martingale property $\mathbb{E}_j\left[G^{(j)} - \nabla f(X^{(j)}) \mid X^{(j)}\right] = 0$ for all $j \leq t$. Next, we use Lemma 13 to estimate the norm $\left\|(k-j)\tilde{W}^{k-j} - (k-j-1)\tilde{W}^{k-j-1}\right\|_2^2 \leq \frac{16}{c^2}$; and using (10) we estimate $\left\|(k-1)\tilde{W}^{k-1}\right\|_2^2 \leq \left\|J^{k-1}\right\|^2 \leq \frac{1}{2}$ due to our choice of $k \geq \tau$ and a key Lemma 4

$$
T_1 \overset{(19)}{\leq} \frac{32k}{c^2}\sum_{j=0}^{k-1}\left\|D^{(t+j)}\right\|_F^2 + \frac{32}{c^2}\sum_{j=0}^{k-1}\left\|Z^{(t+j)}\right\|_F^2
$$

Taking expectation over the stochastic noise,

$$
\mathbb{E}[T_1] \overset{(6)}{\leq} \frac{32k}{c^2}\sum_{j=0}^{k-1}\left\|D^{(t+j)}\right\|_F^2 + \frac{32kn\sigma^2}{c^2}
$$

Summing up $T_1$ and $T_2$ and estimating $k \leq 2\tau$ we conclude the proof

$$
\mathbb{E}\left\|\Psi_{t+k}\right\|_F^2 \leq \frac{3}{4}\mathbb{E}\left\|\Psi_t\right\|_F^2 + \gamma^2\frac{440\tau}{c^2}\sum_{j=0}^{k}\mathbb{E}\left\|\nabla f(X^{(t+j)}) - \nabla f(X^\star)\right\|_F^2 + \gamma^2\frac{380\tau}{c^2}n\sigma^2\,. \quad \square
$$

We will proof Lemma 7 with $B_1 = 28C_1$, $B_2 = 4C_2$, $B_3 = \sqrt{515 \cdot 2C_1}$, where $C_1 = 220$ and $C_2 = 190$ are constants from Lemma 20.

**Proof of Lemma 7.** Observe, for any $t$,

$$
\left\| \nabla f(X^{(t)}) - \nabla f(X^\star) \right\|_F^2 \overset{(18)}{\le} 2 \left\| \nabla f(X^{(t)}) - \nabla f(\bar{X}^{(t)}) \right\|_F^2 + 2 \left\| \nabla f(\bar{X}^{(t)}) - \nabla f(X^\star) \right\|_F^2
$$

$$
\overset{(4)}{\le} 2L^2 \underbrace{\left\| X^{(t)} - \bar{X}^{(t)} \right\|_F^2}_{\le \|\Psi_t\|_F^2} + 2 \left\| \nabla f(\bar{X}^{(t)}) - \nabla f(X^\star) \right\|_F^2
$$

With Lemma 20

$$
\mathbb{E} \left\| \Psi_{t+k} \right\|_F^2 \overset{(20)}{\le} \frac{3}{4} \mathbb{E} \left\| \Psi_t \right\|_F^2 + \gamma^2 \frac{\tau C_1}{c^2} \sum_{j=0}^{k} \mathbb{E} \left\| \nabla f(X^{(t+j)}) - \nabla f(X^\star) \right\|_F^2 + \gamma^2 \frac{\tau C_2}{c^2} n \sigma^2
$$

$$
\le \frac{3}{4} \mathbb{E} \left\| \Psi_t \right\|_F^2 + \gamma^2 \frac{2C_1 \tau L^2}{c^2} \sum_{j=0}^{k} \mathbb{E} \left\| \Psi_{t+j} \right\|_F^2 + \gamma^2 \frac{2C_1 \tau}{c^2} \sum_{j=0}^{k} \mathbb{E} \left\| \nabla f(\bar{X}^{(t+j)}) - \nabla f(X^\star) \right\|_F^2 + \gamma^2 \frac{\tau C_2}{c^2} n \sigma^2
$$

$$
\overset{\gamma < \frac{c}{\sqrt{512 \cdot 2C_1 L \tau}}}{\le} \frac{3}{4} \mathbb{E} \left\| \Psi_t \right\|_F^2 + \frac{1}{512\tau} \sum_{j=0}^{k} \left\| \Psi_{t+j} \right\|_F^2 + \gamma^2 \frac{2C_1 \tau}{c^2} \sum_{j=0}^{k} \mathbb{E} \left\| \nabla f(\bar{X}^{(t+j)}) - \nabla f(X^\star) \right\|_F^2 + \gamma^2 \frac{\tau C_2}{c^2} n \sigma^2
$$

Next, we estimate the third term by smoothness for $j < k$

$$
\left\| \nabla f(\bar{X}^{(t+j)}) - \nabla f(X^\star) \right\|_F^2 \le 2Ln \left( f(\bar{\mathbf{x}}^{(t+j)}) - f(\mathbf{x}^\star) \right) .
$$

And for $j = k$, the index is $t + k$ and it should appear only in LHS. Thus we estimate

$$
\left\| \nabla f(\bar{X}^{(t+k)}) - \nabla f(X^\star) \right\|_F^2 \overset{(18)}{\le} 2 \left\| \nabla f(\bar{X}^{(t+k)}) - \nabla f(\bar{X}^{(t+k-1)}) \right\|_F^2 + 2 \left\| \nabla f(\bar{X}^{(t+k-1)}) - \nabla f(X^\star) \right\|_F^2
$$

$$
\overset{(4)}{\le} 2L^2 \left\| \bar{X}^{(t+k)} - \bar{X}^{(t+k-1)} \right\|_F^2 + 4Ln \left( f(\bar{\mathbf{x}}^{(t+k-1)}) - f(\mathbf{x}^\star) \right)
$$

Next we use (11), that is equivalent to $\bar{X}^{(t+k)} = \bar{X}^{(t+k-1)} - \gamma \nabla F(X^{(t+k-1)}, \xi^{(t+k-1)}) \frac{\mathbf{1}\mathbf{1}^\top}{n}$. Taking expectation

$$
\mathbb{E} \left\| \bar{X}^{(t+k)} - \bar{X}^{(t+k-1)} \right\|_F^2 \overset{(11),(6)}{\le} \gamma^2 \left\| \nabla f(X^{(t+k-1)}) \frac{\mathbf{1}\mathbf{1}^\top}{n} \right\|_F^2 + \gamma^2 \sigma^2
$$

$$
\le 2\gamma^2 \left\| \nabla f(X^{(t+k-1)}) \frac{\mathbf{1}\mathbf{1}^\top}{n} - \nabla \bar{f}(\bar{X}^{(t+k-1)}) \right\|_F^2 + 2\gamma^2 \left\| \nabla \bar{f}(\bar{X}^{(t+k-1)}) - \nabla \bar{f}(X^\star) \right\|_F^2 + \gamma^2 \sigma^2
$$

$$
\overset{(18),(4)}{\le} 2\gamma^2 L^2 \left\| X^{(t+k-1)} - \bar{X}^{(t+k-1)} \right\|_F^2 + 4\gamma^2 Ln \left( f(\bar{\mathbf{x}}^{(t+k-1)}) - f(\mathbf{x}^\star) \right) + \gamma^2 \sigma^2
$$

where on the second line we used $\nabla f(\bar{X}) \frac{\mathbf{1}\mathbf{1}^\top}{n} = \nabla \bar{f}(\bar{X})$, and $\nabla \bar{f}(X^\star) = 0$. As $\gamma \le \frac{c}{\sqrt{512 \cdot 2C_1 L \tau}}$

$$
\left\| \nabla f(\bar{X}^{(t+k)}) - \nabla f(X^\star) \right\|_F^2 \le L^2 \left\| \Psi_{t+k-1} \right\|_F^2 + 5Ln \left( f(\bar{\mathbf{x}}^{(t+k-1)}) - f(\mathbf{x}^\star) \right)
$$

Coming back to recursion for $\| \Psi_{t+k} \|_F^2$ and using that $\frac{2C_1 L^2 \tau}{c^2} \gamma^2 \le \frac{1}{512\tau}$ by our choice of $\gamma$,

$$
\mathbb{E} \left\| \Psi_{t+k} \right\|_F^2 \le \frac{3}{4} \mathbb{E} \left\| \Psi_t \right\|_F^2 + \frac{1}{256\tau} \sum_{j=0}^{k} \left\| \Psi_{t+j} \right\|_F^2 + \gamma^2 \frac{C_1 \tau}{c^2} 14Ln \sum_{j=0}^{k-1} \mathbb{E} \left( f(\bar{\mathbf{x}}^{(t+j)}) - f(\mathbf{x}^\star) \right) + \gamma^2 \frac{2C_2 \tau}{c^2} n \sigma^2
$$

It is only left to get rid of $\| \Psi_{t+k} \|_F^2$ from RHS. For that we move the term with $\| \Psi_{t+k} \|_F^2$ to LHS and divide the whole equation by $\left( 1 - \frac{1}{256\tau} \right)$. We use that $\left( 1 - \frac{1}{256\tau} \right)^{-1} \le 1 + \frac{1}{128\tau} \le 1 + \frac{1}{256} < 2$, and that $\left( 1 - \frac{1}{4} \right) \left( 1 + \frac{1}{128\tau} \right) \le \left( 1 - \frac{1}{4} \right) \left( 1 + \frac{1}{128} \right) \le \left( 1 - \frac{1}{8} \right)$. We thus arrive to the Lemma's statement

$$
\mathbb{E} \left\| \Psi_{t+k} \right\|_F^2 \le \frac{7}{8} \left\| \Psi_t \right\|_F^2 + \frac{1}{128\tau} \sum_{j=0}^{k-1} \mathbb{E} \left\| \Psi_{t+j} \right\|_F^2 + \gamma^2 \frac{28C_1 \tau}{c^2} Ln \sum_{j=0}^{k-1} \mathbb{E} \left( f(\bar{\mathbf{x}}^{(t+j)}) - f(\mathbf{x}^\star) \right) + \gamma^2 \frac{4C_2 \tau}{c^2} n \sigma^2
$$

$\square$

**Proof of Lemma 8.**

*Proof.* Define $\alpha = 28C_1 \frac{\tau}{c^2} Ln$, $\beta = 4C_2 \frac{\tau}{c^2}\sigma^2 n$ for simplicity. Then inequality (14) takes the form

$$\mathbb{E} \|\Psi_{t+k}\|_F^2 \leq \left(1 - \frac{1}{8}\right) \mathbb{E} \|\Psi_t\|_F^2 + \frac{1}{128\tau} \sum_{j=0}^{k-1} \mathbb{E} \|\Psi_{t+j}\|_F^2 + \alpha\gamma^2 \sum_{j=0}^{k-1} \mathbb{E} e_{t+j} + \beta\gamma^2 \qquad (21)$$

**A new quantity.** We define a new quantity that has non-increasing properties even for $k < \tau$ in contrast to $\mathbb{E} \|\Psi_{t+k}\|_F^2$. For $t \geq 0$ we define

$$\Phi_{t+\tau} := \frac{1}{\tau} \sum_{j=0}^{\tau-1} \mathbb{E} \|\Psi_{t+j}\|_F^2 \qquad\qquad E_{t+\tau} := \alpha \sum_{j=0}^{\tau-1} \mathbb{E} e_{t+j}$$

*Non-increasing property for $k < \tau$ (but $t + k \geq \tau$).*

$$\Phi_{t+k} = \frac{1}{\tau} \left( \sum_{i=k}^{\tau-1} \mathbb{E} \|\Psi_{t-\tau+i}\|_F^2 + \sum_{i=0}^{k-1} \mathbb{E} \|\Psi_{t+i}\|_F^2 \right)$$

Applying (21) to the second sum,

$$\Phi_{t+k} \leq \frac{1}{\tau} \sum_{i=k}^{\tau-1} \mathbb{E} \|\Psi_{t-\tau+i}\|_F^2 + \frac{1}{\tau} \sum_{i=0}^{k-1} \left[\left(1 - \frac{1}{8}\right) \mathbb{E} \|\Psi_{t-\tau+i}\|_F^2 + \frac{1}{128}\Phi_{t+i} + \gamma^2 E_{t+i} + \beta\gamma^2 \right]$$

$$\Phi_{t+k} \leq \Phi_t + \frac{1}{128\tau} \sum_{i=0}^{k-1} \Phi_{t+i} + \frac{1}{\tau}\gamma^2 \sum_{i=0}^{k-1} E_{t+i} + \frac{k}{\tau}\beta\gamma^2, \qquad (22)$$

where we used that $\Theta_t \geq 0 \;\forall t$ and that $\tau \geq k$.

*Contraction property for $\tau \leq k \leq 2\tau$.* Using (21) and a definition of $\Phi_{t+k}$,

$$\Phi_{t+k} = \frac{1}{\tau} \sum_{j=k-\tau}^{k-1} \mathbb{E} \|\Psi_{t+j}\|_F^2 \leq \left(1 - \frac{1}{8}\right) \underbrace{\frac{1}{\tau} \sum_{j=k-\tau}^{k-1} \mathbb{E} \|\Psi_{t+j-\tau}\|_F^2}_{\Phi_{t+k-\tau}} + \frac{1}{128\tau} \sum_{j=k-\tau}^{k-1} \Phi_{t+j} + \gamma^2 \frac{1}{\tau} \sum_{i=k-\tau}^{k-1} E_{t+i} + \beta\gamma^2$$

Combining with (22) we get contraction for $\Phi_{t+k}$

$$\Phi_{t+k} \overset{(22)}{\leq} \left(1 - \frac{1}{8}\right) \Phi_t + \frac{1}{128\tau} \sum_{j=0}^{k-1} \Phi_{t+j} + \gamma^2 \frac{1}{\tau} \sum_{i=0}^{k-1} E_{t+i} + 2\beta\gamma^2 \qquad (23)$$

*Simplifying contraction property.* First, we substitute (22) into the second term of (23)

$$\Phi_{t+k} \leq \left(1 - \frac{1}{8}\right) \Phi_t + \frac{1}{128\tau} \sum_{i=0}^{k-2} \Phi_{t+i} + \frac{1}{128\tau} \left[\Phi_t + \frac{1}{128\tau} \sum_{i=0}^{k-2} \Phi_{t+i} + \gamma^2 \frac{1}{\tau} \sum_{i=0}^{k-2} E_{t+i} + 2\beta\gamma^2\right] + \gamma^2 \frac{1}{\tau} \sum_{i=0}^{k-1} E_{t+i} + 2\beta\gamma^2$$

$$\leq \left(1 - \frac{1}{8}\right)\left(1 + \frac{1}{64\tau}\right) \Phi_t + \left(1 + \frac{1}{128\tau}\right)\left[\frac{1}{128\tau} \sum_{i=0}^{k-2} \Phi_{t+i} + \gamma^2 \frac{1}{\tau} \sum_{i=0}^{k-2} E_{t+i} + 2\beta\gamma^2\right] + \gamma^2 \frac{1}{\tau} E_{t+k-1}$$

where we used that $\frac{1}{128\tau} = \left(1 - \frac{1}{2}\right)\frac{1}{64\tau} \leq \left(1 - \frac{1}{8}\right)\frac{1}{64\tau}$. Similarly applying (22) to the rest of $\Phi_{t+i}$,

$$\Phi_{t+k} \leq \left(1 - \frac{1}{8}\right)\left(1 + \frac{1}{64\tau}\right)^k \Phi_t + \gamma^2 \frac{1}{\tau} \sum_{i=0}^{k-1} \left(1 + \frac{1}{128\tau}\right)^{t+k-1-i} E_{t+i} + \left(1 + \frac{1}{128\tau}\right)^k 2\beta\gamma^2$$

We further use $\left(1 + \frac{1}{64\tau}\right)^k \leq \left(1 + \frac{1}{64\tau}\right)^{2\tau} \leq \exp(\frac{1}{32}) \leq 1 + \frac{1}{16}$ and $\left(1 - \frac{1}{8}\right)\left(1 + \frac{1}{64\tau}\right)^k \leq \left(1 - \frac{1}{16}\right)$; and that $\left(1 + \frac{1}{128\tau}\right)^k \leq 1 + \frac{1}{32} \leq 2$. Therefore,

$$\Phi_{t+k} \leq \left(1 - \frac{1}{16}\right) \Phi_t + 2\gamma^2 \frac{1}{\tau} \sum_{i=0}^{k-1} E_{t+i} + 4\beta\gamma^2 \qquad (24)$$

*Simplifying non-increasing property* (22). Similarly as above we substitute recursively (22) into the second term of (22), for $0 < k < \tau$

$$\Phi_{t+k} \le \left(1 + \frac{1}{128\tau}\right)\Phi_t + \left(1 + \frac{1}{128\tau}\right)\left[\frac{1}{128\tau}\sum_{i=0}^{k-2}\Phi_{t+i} + \gamma^2\frac{1}{\tau}\sum_{i=0}^{k-2}E_{t+i} + \beta\gamma^2\right] + \gamma^2\frac{1}{\tau}E_{t+\tau-1}$$

$$\le \left(1 + \frac{1}{128\tau}\right)^{\tau}\Phi_t + \gamma^2\frac{1}{\tau}\sum_{i=0}^{\tau-1}\left(1 + \frac{1}{128\tau}\right)^{t+\tau-1-i}E_{t+i} + \left(1 + \frac{1}{128\tau}\right)^{\tau}\beta\gamma^2$$

Using now that $\left(1 + \frac{1}{128\tau}\right)^{\tau} \le 2$ we get

$$\Phi_{t+k} \le 2\Phi_t + 2\gamma^2\frac{1}{\tau}\sum_{i=0}^{\tau-1}E_{t+i} + 2\beta\gamma^2 \tag{25}$$

**Obtaining recursion for $\mathbb{E}\|\Psi_t\|_F^2 + \Phi_t$.** As our final goal is to obtain inequality for $\mathbb{E}\|\Psi_t\|_F^2$, we start modifying (21), for $\tau \le k \le 2\tau$

$$\mathbb{E}\|\Psi_{t+k}\|_F^2 \le \left(1 - \frac{1}{8}\right)\mathbb{E}\|\Psi_t\|_F^2 + \frac{1}{128}\left(\Phi_{t+k} + \Phi_{t+\tau}\right) + \alpha\gamma^2\sum_{j=0}^{k-1}e_{t+j} + \beta\gamma^2$$

$$\overset{(25)}{\le} \left(1 - \frac{1}{8}\right)\mathbb{E}\|\Psi_t\|_F^2 + \frac{1}{128}\left[4\Phi_t + 4\gamma^2\frac{1}{\tau}\sum_{j=0}^{k-1}E_{t+j}\right] + \alpha\gamma^2\sum_{j=0}^{k-1}e_{t+j} + 2\beta\gamma^2$$

$$\le \left(1 - \frac{1}{8}\right)\mathbb{E}\|\Psi_t\|_F^2 + \frac{1}{32}\Phi_t + \frac{\gamma^2}{32}\frac{1}{\tau}\sum_{j=0}^{k-1}E_{t+j} + \alpha\gamma^2\sum_{j=0}^{k-1}e_{t+j} + 2\beta\gamma^2$$

Summing up the last inequality and (24) we get

$$\mathbb{E}\|\Psi_{t+k}\|_F^2 + \Phi_{t+k} \le \left(1 - \frac{1}{32}\right)\left[\mathbb{E}\|\Psi_t\|_F^2 + \Phi_t\right] + 3\gamma^2\frac{1}{\tau}\sum_{j=0}^{k-1}E_{t+j} + \gamma^2\alpha\sum_{j=0}^{k-1}e_{t+j} + 6\beta\gamma^2$$

*Unrolling recursion up to $\tau$.* For a given $t \ge \tau$, lets define $m = \lfloor t/\tau\rfloor - 1$. Then

$$\mathbb{E}\|\Psi_t\|_F^2 + \Phi_t \le \left(1 - \frac{1}{32}\right)\left[\mathbb{E}\|\Psi_{m\tau}\|_F^2 + \Phi_{m\tau}\right] + 3\gamma^2\frac{1}{\tau}\sum_{j=m\tau}^{t-1}E_j + \gamma^2\alpha\sum_{j=m\tau}^{t-1}e_j + 6\beta\gamma^2$$

Unrolling this recursively up to $\tau$ we get,

$$\mathbb{E}\|\Psi_t\|_F^2 + \Phi_t \le \left(1 - \frac{1}{32}\right)^{m}\left[\mathbb{E}\|\Psi_\tau\|_F^2 + \Phi_\tau\right] + \gamma^2\sum_{j=\tau}^{t-1}\left(1 - \frac{1}{32}\right)^{\lfloor(t-j)/\tau\rfloor}\left[3\frac{1}{\tau}E_j + \alpha e_j\right] + 6\beta\gamma^2\sum_{j=0}^{m-1}\left(1 - \frac{1}{32}\right)^{j}$$

$$\tag{26}$$

**Initial conditions.** Inequality above work for $t \ge \tau$. Here, we focus on $t < \tau$. Using similar calculations as in Lemma 7 replacing estimation of $\left\|\Psi^0 J^t\right\|_F^2$ by Lemma 15, we get that

$$\mathbb{E}\|\Psi_t\|_F^2 \le \underbrace{2\left\|\Delta X^{(0)}\right\|_F^2 + \frac{3\gamma^2}{p^2}\left\|\Delta Y^{(0)}\right\|_F^2}_{:=\tilde{\Theta}_0} + \frac{1}{128\tau}\sum_{j=0}^{t-1}\mathbb{E}\|\Psi_j\|_F^2 + \alpha\gamma^2\sum_{j=0}^{t-1}e_j + \beta\gamma^2 \tag{27}$$

Recursively applying (27) to the second term of (27), similarly as above, we get

$$\mathbb{E}\|\Psi_t\|_F^2 \le 2\tilde{\Theta}_0 + 2\alpha\gamma^2\frac{1}{\tau}\sum_{j=0}^{t-1}e_j + 2\beta\gamma^2 \tag{28}$$

And therefore,

$$\Phi_\tau = \frac{1}{\tau}\sum_{j=0}^{\tau-1}\mathbb{E}\|\Psi_j\|_F^2 \le 2\tilde{\Theta}_0 + 2\alpha\gamma^2\frac{1}{\tau}\sum_{j=0}^{\tau-1}e_j + 2\beta\gamma^2 \tag{29}$$

**Final recursion.** Finally we apply (28), (29) to the first term of (26)

$$\mathbb{E}\left\|\Psi_t\right\|_F^2 + \Phi_t \leq \left(1 - \frac{1}{32}\right)^m 4\tilde{\Theta}_0 + \gamma^2 \sum_{j=\tau}^{t-1} \left(1 - \frac{1}{32}\right)^{\lfloor(t-j)/\tau\rfloor} \left[3\frac{1}{\tau}E_j + 5\alpha e_j\right] + 10\beta\gamma^2 \sum_{j=0}^{m-1}\left(1 - \frac{1}{32}\right)^j$$

- For the last term we estimate $\sum_{j=0}^{m-1}\left(1 - \frac{1}{32}\right)^j \leq 2$.
- For the terms with $e_j$ and $E_j$ we estimate, similar to [16],

$$\left(1 - \frac{1}{32}\right)^{1/\tau} \leq \exp(-\frac{1}{32\tau}) \leq 1 - \frac{1}{64\tau} \qquad \text{and thus}$$

$$\left(1 - \frac{1}{32}\right)^{\lfloor(t-j)/\tau\rfloor} \leq \left(1 - \frac{1}{64\tau}\right)^{\tau\lfloor(t-j)/\tau\rfloor} \leq \left(1 - \frac{1}{64\tau}\right)^{t-j}\left(1 - \frac{1}{64\tau}\right)^{-\tau} \leq 2\left(1 - \frac{1}{64\tau}\right)^{t-j}$$

  where as $\frac{1}{64\tau} \leq \frac{1}{2}$ we estimated $\left(1 - \frac{1}{64\tau}\right)^{-\tau} \leq \left(\frac{1}{1-\frac{1}{64\tau}}\right)^\tau \leq (1 + \frac{1}{32\tau})^\tau \leq \exp(\frac{1}{32}) < 2$.

- Similarly, for $\tilde{\Theta}_0$ term we estimate
  $\left(1 - \frac{1}{32}\right)^m = \left(1 - \frac{1}{32}\right)^{\lfloor\frac{t-\tau}{\tau}\rfloor} \leq \left(1 - \frac{1}{64\tau}\right)^t\left(1 - \frac{1}{64\tau}\right)^{-2\tau} \leq 2\left(1 - \frac{1}{64\tau}\right)^t$.

- For the terms with $E_j$ we additionally estimate

$$2\left(1 - \frac{1}{64\tau}\right)^{t-j} E_j = 2\left(1 - \frac{1}{64\tau}\right)^{t-j}\sum_{i=j-\tau}^{j-1} e_i = 2\sum_{i=j-\tau}^{j-1}\left(1 - \frac{1}{64\tau}\right)^{t-i}\left(1 - \frac{1}{64\tau}\right)^{i-j} e_i$$

  Further, $-\tau < i - j < -1$, and thus $\left(1 - \frac{1}{64\tau}\right)^{i-j} \leq 2$ for all such $-\tau < i - j < -1$.

Therefore we obtain

$$\mathbb{E}\left\|\Psi_t\right\|_F^2 + \Phi_t \leq \left(1 - \frac{1}{64\tau}\right)^t 8\tilde{\Theta}_0 + 22\gamma^2\alpha\sum_{j=0}^{t-1}\left(1 - \frac{1}{64\tau}\right)^{t-j} e_j + 20\beta\gamma^2$$

This brings us to the statement of the lemma. $\qquad\square$

The rest of the proof follows closely [16].

### B.2.1 $\tau$-slow Sequences

**Definition 2** ($\tau$-slow sequences [44])**.** *The sequence $\{a_t\}_{t\geq 0}$ of positive values is $\tau$-slow decreasing for parameter $\tau > 0$ if*

$$a_{t+1} \leq a_t, \quad \forall t \geq 0 \qquad \text{and,} \qquad a_{t+1}\left(1 + \frac{1}{2\tau}\right) \geq a_t, \quad \forall t \geq 0.$$

*The sequence $\{a_t\}_{t\geq 0}$ is $\tau$-slow increasing if $\{a_t^{-1}\}_{t\geq 0}$ is $\tau$-slow decreasing.*

**Proposition 21** (Examples)**.**

1. *The sequence $\{\eta_t^2\}_{t\geq 0}$ with $\eta_t = \frac{a}{b+t}$, $b \geq 32\tau$ is $4\tau$-slow decreasing.*

2. *The sequence of constant stepsizes $\{\eta_t^2\}_{t\geq 0}$ with $\eta_t = \eta$ is $\tau$-slow decreasing for any $\tau$.*

3. *The sequence $\{w_t\}_{t\geq 0}$ with $w_t = (b + t)^2$, $b \geq 84\tau$ is $8\tau$-slow increasing.*

4. *The sequence of constant weights $\{w_t\}_{t\geq 0}$ with $w_t = 1$ is $\tau$-slow increasing for any $\tau$.*

### B.2.2 The Main Recursion

**Lemma 22** (The main recursion). *Let $\{w_t\}_{t \geq 0}$ be $64\tau$-slow increasing sequence, $W_t = \frac{1}{T+1} \sum_{t=0}^{T} w_t$, with $\gamma \leq \frac{c}{582 C_1 \tau L}$ it holds that*

$$\sum_{t=0}^{T} w_t \, \mathbb{E} \left\| \Psi_t \right\|_F^2 \leq \sum_{t=0}^{T} w_t \left( 1 - \frac{1}{64\tau} \right)^t 8 \tilde{\Theta}_0 + \frac{n}{6L} \sum_{t=0}^{T} e_t w_t + 40 C_2 \frac{\tau}{c^2} \sigma^2 n \gamma^2 W_T, \qquad (30)$$

*where $e_t = f(\bar{\mathbf{x}}^{(t)}) - f^\star$, $\tilde{\Theta}_0 = 2 \left\| \Delta X^{(0)} \right\|_F^2 + \frac{3\gamma^2}{p^2} \left\| \Delta Y^{(0)} \right\|_F^2$, $C_1 = 440, C_2 = 380$.*

*Proof.* We start by averaging (15) with weights $w_t$. Define $W_T = \sum_{t=0}^{T} w_t$, $\alpha = 28 C_1 \frac{\tau}{c^2} Ln$, $\beta = 4 C_2 \frac{\tau}{c^2} \sigma^2 n$,

$$\sum_{t=0}^{T} w_t \, \mathbb{E} \left\| \Psi_t \right\|_F^2 \leq \sum_{t=0}^{T} w_t \left( 1 - \frac{1}{64\tau} \right)^t 8 \tilde{\Theta}_0 + 22 \gamma^2 \alpha \underbrace{\sum_{t=0}^{T} w_t \sum_{j=0}^{t-1} \left( 1 - \frac{1}{64\tau} \right)^{t-j} e_j}_{:=T_1} + 20 \beta \gamma^2 W_T$$

For the middle term $T_1$ we use that $w_t$ are $64\tau$-slow increasing sequences, i.e. $w_t \leq w_j \left( 1 + \frac{1}{128\tau} \right)^{t-j}$, we get

$$T_1 = \sum_{t=0}^{T} \sum_{j=0}^{t-1} \left( 1 - \frac{1}{64\tau} \right)^{t-j} \left( 1 + \frac{1}{128\tau} \right)^{t-j} e_j w_j \leq \sum_{t=0}^{T} \sum_{j=0}^{t-1} \left( 1 - \frac{1}{128\tau} \right)^{t-j} e_j w_j$$

$$\leq \sum_{j=0}^{T} e_j w_j \sum_{t=j+1}^{T} \left( 1 - \frac{1}{128\tau} \right)^{t-j} \leq \sum_{j=0}^{\infty} e_j w_j \sum_{t=0}^{\infty} \left( 1 - \frac{1}{128\tau} \right)^{t-j} \leq 128\tau \sum_{t=0}^{T} e_t w_t$$

Therefore,

$$\sum_{t=0}^{T} w_t \Theta_t \leq \sum_{t=0}^{T} w_t \left( 1 - \frac{1}{64\tau} \right)^t 8 \tilde{\Theta}_0 + 2816 \gamma^2 \alpha \tau \sum_{t=0}^{T} e_t w_t + 20 \beta \gamma^2 W_T$$

Now using that $\gamma \leq \frac{c}{582 C_1 \tau L}$ and that $\alpha = 20 C_1 \frac{\tau}{c^2} Ln$, $\beta = 2 C_2 \frac{\tau}{c^2} \sigma^2 n$.

$$\sum_{t=0}^{T} w_t \Theta_t \leq \sum_{t=0}^{T} w_t \left( 1 - \frac{1}{64\tau} \right)^t 8 \tilde{\Theta}_0 + \frac{n}{6L} \sum_{t=0}^{T} e_t w_t + 40 C_2 \frac{\tau}{c^2} \sigma^2 n \gamma^2 W_T$$

$\square$

### B.2.3 Combining with the Descent Lemma 6

**Lemma 23.** *Define $D = \frac{\sigma^2}{n}$, $a = \frac{\mu}{2}$, $A = 24 L \frac{1}{n} \tilde{\Theta}_0$, $\tilde{\Theta}_0 = 2 \left\| \Delta X^{(0)} \right\|_F^2 + \frac{3\gamma^2}{p^2} \left\| \Delta Y^{(0)} \right\|_F^2$, $B = 120 C_2 L \frac{\tau}{c^2} \sigma^2$, $C_1 = 440, C_2 = 380$. Then with $\gamma \leq \frac{c}{582 C_1 \tau L}$ it holds that*

$$\frac{1}{2 W_T} \sum_{t=0}^{T} w_t e_t \leq \frac{1}{W_T} \sum_{t=0}^{T} \left( \frac{(1 - \gamma a)}{\gamma} w_t r_t - \frac{w_t}{\gamma} r_{t+1} \right) + D\gamma + \frac{A}{W_T} \sum_{t=0}^{T} w_t \left( 1 - \frac{1}{64\tau} \right)^t + B\gamma^2 \tag{31}$$

*Proof.* First, define $W_T = \sum_{t=0}^{T} w_t$, $r_t = \left\| \bar{\mathbf{x}}^{(t)} - \mathbf{x}^\star \right\|^2$, $\Theta_t = \sum_{i=1}^{n} \mathbb{E} \left\| \bar{\mathbf{x}}^{(t)} - \mathbf{x}_i^{(t)} \right\|^2$. In this notation, (12) writes as

$$r_{t+1} \leq \left( 1 - \frac{\gamma\mu}{2} \right) r_t + \frac{\gamma^2 \sigma^2}{n} - \gamma e_t + \gamma \frac{3L}{n} \Theta_t,$$

We rearrange (12) by multiplying by $w_t$ and dividing by $\gamma$

$$w_t e_t \leq \frac{\left( 1 - \frac{\gamma\mu}{2} \right)}{\gamma} w_t r_t - \frac{w_t}{\gamma} r_{t+1} + \frac{\sigma^2}{n} w_t \gamma + \frac{3L}{n} w_t \Theta_t,$$

Now summing up, dividing by $W_T$, using that $\Theta_t \leq \mathbb{E}\left\|\Psi_t\right\|_F^2$, and using (30)

$$\frac{1}{W_T}\sum_{t=0}^{T}w_t e_t \leq \frac{1}{W_T}\sum_{t=0}^{T}\left(\frac{\left(1-\frac{\gamma\mu}{2}\right)}{\gamma}w_t r_t - \frac{w_t}{\gamma}r_{t+1}\right) + \frac{\sigma^2}{n}\gamma + \frac{1}{W_T}\sum_{t=0}^{T}w_t 24L\left(1-\frac{1}{64\tau}\right)^t\frac{1}{n}\tilde{\Theta}_0$$

$$+ \frac{1}{2}\frac{1}{W_T}\sum_{t=0}^{T}e_t w_t + 120C_2 L\frac{\tau}{c^2}\sigma^2\gamma^2$$

Putting the fourth term to LHS we get the statement of the lemma. $\qquad\square$

Now similar to [16, Lemma 15] we obtain the rates of Theorem 2 for the strongly convex case, and similar to [16, Lemma 16] for the weakly convex case.

### B.2.4 Strongly Convex Case

**Lemma 24.** *If non-negative sequences* $\{r_t\}_{t\geq 0}, \{e_t\}_{t\geq 0}$ *satisfy* (31) *for some constants* $a > 0$, $D, A, B \geq 0$, *then there exists a constant stepsize* $\gamma < \frac{1}{b}$ *with* $b \geq 128a\tau$ *such that for weights* $w_t = (1-a\gamma)^{-(t+1)}$ *and* $W_T := \sum_{t=0}^{T}w_t$ *it holds:*

$$\frac{1}{2W_T}\sum_{t=0}^{T}e_t w_t + ar_{T+1} \leq \tilde{\mathcal{O}}\left((r_0 + {}^{A}\!/\!{}_{2a})b\exp\left[-\frac{a(T+1)}{b}\right] + \frac{D}{aT} + \frac{B}{a^2T^2}\right),$$

*where* $\tilde{\mathcal{O}}$ *hides polylogarithmic factors.*

*Proof.* Starting from (31) and using that that $\frac{w_t(1-a\gamma)}{\gamma} = \frac{w_{t-1}}{\gamma}$ we obtain a telescoping sum,

$$\frac{1}{2W_T}\sum_{t=0}^{T}w_t e_t \leq \frac{1}{W_T\gamma}\left((1-a\gamma)w_0 r_0 - w_T r_{T+1}\right) + D\gamma + B\gamma^2 + \frac{A}{W_T}\sum_{t=0}^{T}w_t\left(1-\frac{1}{64\tau}\right)^t,$$

And hence,

$$\frac{1}{2W_T}\sum_{t=0}^{T}w_t e_t + \frac{w_T r_{T+1}}{W_T\gamma} \leq \frac{r_0}{W_T\gamma} + D\gamma + B\gamma^2 + \frac{A}{W_T}\sum_{t=0}^{T}w_t\left(1-\frac{1}{64\tau}\right)^t,$$

Now we estimate the last term. We use that $2\gamma a \leq \frac{1}{64\tau}$ and thus $\left(1-\frac{1}{64\tau}\right)^t \leq (1-a\gamma)^{2t}$

$$\frac{1}{W_T}\sum_{t=0}^{T}(1-a\gamma)^{-(t+1)}\left(1-\frac{1}{64\tau}\right)^t \leq \frac{1}{W_T}\sum_{t=0}^{T}(1-a\gamma)^{t-1} \leq \frac{1}{W_T}\frac{1}{2a\gamma}$$

where we used that $\frac{1}{1-a\gamma} \leq \frac{1}{2}$. Thus,

$$\frac{1}{2W_T}\sum_{t=0}^{T}w_t e_t + \frac{w_T r_{T+1}}{W_T\gamma} \leq \frac{1}{W_T\gamma}\left(r_0 + \frac{A}{2a}\right) + D\gamma + B\gamma^2,$$

Using that $W_T \leq \frac{w_T}{a\gamma}$ and $W_T \geq w_T = (1-a\gamma)^{-(T+1)}$ we can simplify

$$\frac{1}{2W_T}\sum_{t=0}^{T}w_t e_t + ar_{T+1} \leq (1-a\gamma)^{T+1}\frac{1}{\gamma}\left(r_0 + \frac{A}{2a}\right) + D\gamma + B\gamma^2 \leq \frac{1}{\gamma}\left(r_0 + \frac{A}{2a}\right)\exp\left[-a\gamma(T+1)\right] + D\gamma + B\gamma^2,$$

Now lemma follows by tuning $\gamma$ the same way as in [43].

- If $\frac{1}{b} \geq \frac{\ln(\max\{2, a^2(r_0+\frac{A}{2a})T^2/D\})}{aT}$ then we choose $\eta = \frac{\ln(\max\{2, a^2(r_0+\frac{A}{2a})T^2/D\})}{aT}$ and get that

$$\tilde{\mathcal{O}}\left(a(r_0 + {}^{A}\!/\!{}_{2a})T\exp\left[-\ln(\max\{2, a^2(r_0 + {}^{A}\!/\!{}_{2a})T^2/D\})\right]\right) + \tilde{\mathcal{O}}\left(\frac{D}{aT}\right) + \tilde{\mathcal{O}}\left(\frac{B}{a^2T^2}\right)$$

$$= \tilde{\mathcal{O}}\left(\frac{D}{aT}\right) + \tilde{\mathcal{O}}\left(\frac{B}{a^2T^2}\right),$$

- Otherwise $\frac{1}{b} \leq \frac{\ln(\max\{2, a^2 (r_0 + \frac{A}{2a}) T^2 / D\})}{aT}$ we pick $\eta = \frac{1}{b}$ and get that

$$\tilde{\mathcal{O}}\left( (r_0 + A/2a) b \exp\left[-\frac{a(T+1)}{b}\right] + \frac{D}{b} + \frac{B}{b^2}\right)$$

$$\leq \tilde{\mathcal{O}}\left( (r_0 + A/2a) b \exp\left[-\frac{a(T+1)}{b}\right] + \frac{D}{aT} + \frac{B}{a^2 T^2}\right). \qquad \square$$

## B.3 Weakly Convex and Non Convex Cases

**Lemma 25.** *If non-negative sequences $\{r_t\}_{t \geq 0}, \{e_t\}_{t \geq 0}$ satisfy (31) with $a = 0$, $D, A, B \geq 0$, then there exists a constant stepsize $\gamma < \frac{1}{b}$ with $b \geq 128a\tau$ such that for weights $\{w_t = 1\}_{t \geq 0}$ it holds that:*

$$\frac{1}{(T+1)} \sum_{t=0}^{T} e_t \leq \mathcal{O}\left( 2\left(\frac{cr_0}{T+1}\right)^{\frac{1}{2}} + 2B^{1/3}\left(\frac{r_0}{T+1}\right)^{\frac{2}{3}} + \frac{br_0 + A\tau}{T+1}\right).$$

*Proof.* With $a = 0$, constant stepsizes $\eta_t = \eta$ and weights $\{w_t = 1\}_{t \geq 0}$ (31) is equivalent to

$$\frac{1}{2(T+1)} \sum_{t=0}^{T} e_t \leq \frac{1}{(T+1)\gamma} \sum_{t=0}^{T} (r_t - r_{t+1}) + D\gamma + B\gamma^2 + \frac{A}{T+1} \sum_{t=0}^{T} \left(1 - \frac{1}{64\tau}\right)^t$$

$$\leq \frac{r_0}{(T+1)\gamma} + D\gamma + B\gamma^2 + \frac{64A\tau}{T+1}.$$

To conclude the proof we tune the stepsize for the first three terms using Lemma 26. $\qquad \square$

**Lemma 26** (Tuning the stepsize). *For any parameters $r_0 \geq 0, b \geq 0, e \geq 0, d \geq 0$ there exists constant stepsize $\eta \leq \frac{1}{b}$ such that*

$$\Psi_T := \frac{r_0}{\gamma(T+1)} + D\eta + B\eta^2 \leq 2\left(\frac{Dr_0}{T+1}\right)^{\frac{1}{2}} + 2B^{1/3}\left(\frac{r_0}{T+1}\right)^{\frac{2}{3}} + \frac{br_0}{T+1}$$

*Proof.* Choosing $\eta = \min\left\{ \left(\frac{r_0}{D(T+1)}\right)^{\frac{1}{2}}, \left(\frac{r_0}{B(T+1)}\right)^{\frac{1}{3}}, \frac{1}{b}\right\} \leq \frac{1}{b}$ we have three cases

- $\eta = \frac{1}{b}$ and is smaller than both $\left(\frac{r_0}{D(T+1)}\right)^{\frac{1}{2}}$ and $\left(\frac{r_0}{B(T+1)}\right)^{\frac{1}{3}}$, then

$$\Psi_T \leq \frac{br_0}{T+1} + \frac{D}{b} + \frac{B}{b^2} \leq \left(\frac{Dr_0}{T+1}\right)^{\frac{1}{2}} + \frac{br_0}{T+1} + B^{1/3}\left(\frac{r_0}{T+1}\right)^{\frac{2}{3}}$$

- $\eta = \left(\frac{r_0}{D(T+1)}\right)^{\frac{1}{2}} < \left(\frac{r_0}{B(T+1)}\right)^{\frac{1}{3}}$, then

$$\Psi_T \leq 2\left(\frac{r_0 D}{T+1}\right)^{\frac{1}{2}} + B\left(\frac{r_0}{D(T+1)}\right) \leq 2\left(\frac{r_0 D}{T+1}\right)^{\frac{1}{2}} + B^{\frac{1}{3}}\left(\frac{r_0}{(T+1)}\right)^{\frac{2}{3}},$$

- The last case, $\eta = \left(\frac{r_0}{B(T+1)}\right)^{\frac{1}{3}} < \left(\frac{r_0}{D(T+1)}\right)^{\frac{1}{2}}$

$$\Psi_T \leq 2B^{\frac{1}{3}}\left(\frac{r_0}{(T+1)}\right)^{\frac{2}{3}} + D\left(\frac{r_0}{B(T+1)}\right)^{\frac{1}{3}} \leq 2B^{\frac{1}{3}}\left(\frac{r_0}{(T+1)}\right)^{\frac{2}{3}} + \left(\frac{Dr_0}{T+1}\right)^{\frac{1}{2}}. \quad \square$$

## B.4 Non-convex Case

First, we state the descent Lemma for non-convex cases. Due to Lemma 5, it holds that

**Lemma 27** (Descent lemma for non-convex case, Lemma 11 from [16]). *Under Assumptions as in Theorem 2, the averages $\bar{\mathbf{x}}^{(t)} := \frac{1}{n}\sum_{i=1}^{n}\mathbf{x}_i^{(t)}$ of the iterates of Algorithm 1 with the constant stepsize $\gamma < \frac{1}{4L(M+1)}$ satisfy*

$$\mathbb{E}_{t+1}\, f(\bar{\mathbf{x}}^{(t+1)}) \leq f(\bar{\mathbf{x}}^{(t)}) - \frac{\gamma}{4}\left\|\nabla f(\bar{\mathbf{x}}^{(t)})\right\|_2^2 + \frac{\gamma L^2}{n}\sum_{i=1}^{n}\left\|\bar{\mathbf{x}}^{(t)} - \mathbf{x}_i^{(t)}\right\|_2^2 + \frac{L}{n}\gamma^2\sigma^2. \tag{32}$$

Similarly as for the convex cases we prove the following recursion

**Lemma 28** (Consensus distance recursion). *There are exists absolute constants $C_1, C_2 > 0$ such that*

$$\mathbb{E}\left\|\Psi_{t+k}\right\|_F^2 \leq \frac{3}{4}\left\|\Psi_t\right\|_F^2 + \frac{1}{128\tau}\sum_{j=0}^{k-1}\mathbb{E}\left\|\Psi_{t+j}\right\|_F^2 + C_1\gamma^2\tau n\sum_{j=0}^{k-1}e_{t+j} + C_2\gamma^2\left(\frac{\tau n}{c^2} + \tau^2\right)\sigma^2 \tag{33}$$

*where $e_j = \left\|\nabla f(\bar{\mathbf{x}}^{(j)})\right\|^2$, $\tau \leq k \leq 2\tau$, $\tau = \frac{2}{p}\log\left(\frac{50}{p}(1+\log\frac{1}{p})\right) + 1$, $p$ and $c$ are defined in (3), $\Psi_t = \left(\Delta X^{(t)}, \gamma\Delta Y^{(t)}\right)$ and is defined in (9).*

*Proof.* The proof starts exactly the same as in the convex cases, Lemma 20. The difference comes when estimating terms $T_1$ and $T_2$.

**The second term $T_2$.** After splitting the stochastic noise,

$$\mathbb{E}[T_2] \leq 3\,\mathbb{E}\left\|\sum_{j=1}^{k}\left(\nabla f(X^{(t+j)}) - \nabla f(X^{(t+j-1)})\right)\tilde{W}^{\tau-j}\right\|_F^2 + 6kn\sigma^2$$

$$\overset{(16)}{\leq} 3k\sum_{j=1}^{k}\mathbb{E}\left\|\nabla f(X^{(t+j)}) - \nabla f(X^{(t+j-1)})\right\|_F^2 + 6kn\sigma^2$$

Estimating separately

$$\mathbb{E}\left\|\nabla f(X^{(t+j)}) - \nabla f(X^{(t+j-1)})\right\|_F^2 \overset{(16)}{\leq} 3\,\mathbb{E}\left\|\nabla f(X^{(t+j)}) - \nabla f(\bar{X}^{(t+j)})\right\|_F^2 + 3\left\|\nabla f(\bar{X}^{(t+j-1)}) - \nabla f(X^{(t+j-1)})\right\|_F^2$$

$$+ 3\left\|\nabla f(\bar{X}^{(t+j)}) - \nabla f(\bar{X}^{(t+j-1)})\right\|_F^2$$

$$\overset{(4)}{\leq} 3L^2\,\mathbb{E}\left\|X^{(t+j)} - \bar{X}^{(t+j)}\right\|_F^2 + 3L^2\left\|\bar{X}^{(t+j-1)} - X^{(t+j-1)}\right\|_F^2$$

$$+ 3L^2\left\|\bar{X}^{(t+j)} - \bar{X}^{(t+j-1)}\right\|_F^2$$

And for the last term we estimate

$$\mathbb{E}\left\|\bar{\mathbf{x}}^{(t+j)} - \bar{\mathbf{x}}^{(t+j-1)}\right\|_2^2 \leq \gamma^2\left\|\frac{1}{n}\sum_{i=1}^{n}\nabla f_i(\mathbf{x}_i^{(t+j-1)})\right\|_2^2 + \gamma^2\frac{\sigma^2}{n}$$

$$\leq 2\gamma^2\left\|\frac{1}{n}\sum_{i=1}^{n}\nabla f_i(\mathbf{x}_i^{(t+j-1)}) - \frac{1}{n}\sum_{i=1}^{n}\nabla f_i(\bar{\mathbf{x}}^{(t+j-1)})\right\|_2^2 + 2\gamma^2\left\|\nabla f(\bar{\mathbf{x}}^{(t+j-1)})\right\|^2 + \gamma^2\frac{\sigma^2}{n}$$

$$\leq 2\gamma^2 L^2\frac{1}{n}\sum_{i=1}^{n}\left\|\mathbf{x}_i^{(t+j-1)} - \bar{\mathbf{x}}^{(t+j-1)}\right\|^2 + 2\gamma^2\left\|\nabla f(\bar{\mathbf{x}}^{(t+j-1)})\right\|^2 + \gamma^2\frac{\sigma^2}{n}$$

Thus, using that $\gamma < \frac{1}{24L\tau}$, $k \leq 2\tau$

$$\mathbb{E}[T_2] \leq \tau\sum_{j=0}^{k-1}n\,\mathbb{E}\left\|\nabla f(\bar{\mathbf{x}}^{(t+j)})\right\|^2 + 21L^2\tau\sum_{j=0}^{k-1}\mathbb{E}\left\|X^{(t+j)} - \bar{X}^{(t+j)}\right\|_F^2 + 7\tau n\sigma^2.$$

**Term $T_1$.** Similarly, after separating the stochastic noise with $Z^{(t)} = G^{(t)} - \nabla f(X^{(t)})$,

$$T_1 \overset{(18)}{\leq} 2 \left\| \sum_{j=1}^{k} \left[ \nabla f(X^{(t+j)}) - \nabla f(X^{(t+j-1)}) \right] (k-j) \tilde{W}^{k-j} \right\|_F^2 + 2 \left\| \sum_{j=1}^{k} \left( Z^{(t+j)} - Z^{(t+j-1)} \right) (k-j) \tilde{W}^{k-j} \right\|_F^2 .$$

We add and subtract $\nabla f(\bar{X}^{t+j}), \nabla f(\bar{X}^{t+j-1})$ in the first term and denote $D^{(j)} = \nabla f(X^{(j)}) - \nabla f(\bar{X}^{(j)})$.

$$T_1 \leq 4 \left\| \sum_{j=1}^{k} \left( D^{(t+j)} - D^{(t+j-1)} \right) (k-j) \tilde{W}^{k-j} \right\|_F^2 + 4 \left\| \sum_{j=1}^{k} \left[ \nabla f(\bar{X}^{t+j}) - \nabla f(\bar{X}^{t+j-1}) \right] (k-j) \tilde{W}^{k-j} \right\|_F^2$$

$$+ 2 \left\| \sum_{j=1}^{k} \left( Z^{(t+j)} - Z^{(t+j-1)} \right) (k-j) \tilde{W}^{k-j} \right\|_F^2 .$$

Terms with $D$ and $Z$ we estimate exactly the same as in the convex case, thus getting

$$\mathbb{E}[T_1] \overset{(6)}{\leq} \frac{64k}{c^2} \sum_{j=0}^{k-1} \left\| D^{(t+j)} \right\|_F^2 + \frac{32kn\sigma^2}{c^2} + 4 \underbrace{ \left\| \sum_{j=1}^{k} \left[ \nabla f(\bar{X}^{t+j}) - \nabla f(\bar{X}^{t+j-1}) \right] (k-j) \tilde{W}^{k-j} \right\|_F^2 }_{T_3}$$

It is only left to estimate the last term. For that we use Lemma 14, and $\frac{1}{p} \leq \tau$ due to our choice of $\tau$,

$$T_3 \overset{(16)}{\leq} k \sum_{j=1}^{k} \left\| \left[ \nabla f(\bar{X}^{t+j}) - \nabla f(\bar{X}^{t+j-1}) \right] (k-j) \tilde{W}^{k-j} \right\|_F^2 \overset{\text{L. 14}}{\leq} 4k\tau^2 \sum_{j=1}^{k} \left\| \nabla f(\bar{X}^{t+j}) - \nabla f(\bar{X}^{t+j-1}) \right\|_F^2$$

$$\leq 4k\tau^2\gamma^2 \sum_{j=1}^{k} \left[ 2L^2 \left\| X^{(t+j-1)} - \bar{X}^{(t+j-1)} \right\|_F^2 + 2n \left\| \nabla f(\bar{\mathbf{x}}^{(t+j-1)}) \right\|^2 + \sigma^2 \right]$$

Where the last inequality was obtained while estimating Term $T_2$. Using that $k \leq 2\tau$, $\gamma \leq \frac{1}{24L\tau}$ and that $\left\| D^{(t+j)} \right\|_F^2 \leq L^2 \left\| X^{(t+j)} - \bar{X}^{(t+j)} \right\|_F^2$ by smoothness

$$\mathbb{E}[T_1] \overset{(6)}{\leq} \frac{129\tau}{c^2} L^2 \sum_{j=0}^{k-1} \left\| X^{(t+j)} - \bar{X}^{(t+j)} \right\|_F^2 + \tau \sum_{j=0}^{k-1} n \left\| \nabla f(\bar{\mathbf{x}}^{(t+j)}) \right\|^2 + \left( \frac{64\tau n}{c^2} + \tau^2 \right) \sigma^2$$

Summing $T_1$ and $T_2$ together, and using that $\gamma \leq \frac{c}{310\tau L}$

$$\mathbb{E} \left\| \Psi_{t+k} \right\|_F^2 \leq \frac{3}{4} \left\| \Psi_t \right\|_F^2 + \frac{1}{128\tau} \sum_{j=0}^{k-1} \mathbb{E} \left\| \Psi_{t+j} \right\|_F^2 + \gamma^2 10\tau n \sum_{j=0}^{k-1} \left\| \nabla f(\bar{\mathbf{x}}^{(t+j)}) \right\|^2 + 5\gamma^2 \left( \frac{64\tau n}{c^2} + \tau^2 \right) \sigma^2$$

$\square$

Next, we unroll this recursion with Lemma 8.

For $\gamma < \frac{c}{\sqrt{7B_1}L\tau} \leq \frac{1}{2L\tau}$, and with some positive absolute constants $B_1, B_2 > 0$ it holds,

$$\mathbb{E} \left\| \Psi_t \right\|_F^2 \leq \left( 1 - \frac{1}{64\tau} \right)^t A_0 + B_1 \tau \gamma^2 \sum_{j=0}^{t-1} \left( 1 - \frac{1}{64\tau} \right)^{t-j} n e_j + B_2 \gamma^2 \left( \frac{\tau n}{c^2} + \tau^2 \right) \sigma^2 \qquad (34)$$

where $e_j = \left\| \nabla f(\bar{\mathbf{x}}^{(j)}) \right\|^2$, $A_0 = 16 \| \Delta X^{(0)} \|_F^2 + \frac{24\gamma^2}{p^2} \| \Delta Y^{(0)} \|_F^2$.

The rest of proof consists of combining (34) with the descent lemma for non-convex case (32) in similar fashion as in Lemmas 22, 23; and further using Lemma 25 to obtain the final rate.

# C Experimental Setup and Additional Plots

We illustrate the dependence of the convergence rate on the parameters $c$ and $p$.

In these experiments, we vary $p$ and $c$ (by changing the mixing matrix) and measure the value of $f(\bar{\mathbf{x}}^{(t)}) - f^\star$ that GT reaches after a large number of steps $t$, when using a constant stepsize $\gamma$ (chosen small enough so that none of the runs diverges). According to our theoretical results, GT converges to the level $\mathcal{O}\left(\frac{\gamma\sigma^2}{n} + \frac{\gamma^2\sigma^2}{pc^2}\right)$ in a linear number of steps (to reach higher accuracy, smaller stepsizes must be used). Thus, for $n$ large enough, this term is dominated by $\mathcal{O}\left(\frac{\gamma^2\sigma^2}{pc^2}\right)$, which we aim to measure. In all experiments we ensure that the first term is at least by order of magnitude smaller than the second by comparing the noise level with GT on a fully-connected topology.

## C.1 Problem Instances

We used $n = 300$, $d = 100$.

**Setup A (Gaussian Noise).** We consider quadratic functions defined as $f_i(\mathbf{x}) = \|\mathbf{x}\|^2$, and $\mathbf{x}^{(0)}$ is randomly initialized from a normal distribution $\mathcal{N}(0,1)$. We add artificially stochastic noise to gradients as $\nabla F_i(\mathbf{x}, \xi) = \nabla f_i(\mathbf{x}) + \xi$, where $\xi \sim \mathcal{N}(0, \frac{\sigma^2}{d}I)$.

**Setup B (Structured Noise).** We consider quadratic functions defined as $f_i(\mathbf{x}) = \|\mathbf{x}\|^2$, and $\mathbf{x}^{(0)}$ is randomly initialized from a normal distribution $\mathcal{N}(0,1)$. We add artificially stochastic noise to gradients as $\nabla F(X, \xi) = \nabla f(X) + \text{diag}(\xi)V$, where $\xi \sim \mathcal{N}(0, \frac{\sigma^2}{d}I)$ is a $d$-dimensional Gaussian noise vector, $\text{diag}(\xi)$ a matrix with $\xi$ on the diagonal, and $V \in \mathbb{R}^{d \times n}$ is a matrix with half of the rows equal to $\mathbf{v} \in \mathbb{R}^n$, and half of the rows equal to $\mathbf{u} \in \mathbb{R}^n$, where $\mathbf{v}, \mathbf{u}$ are eigenvectors of the mixing matrix, $W\mathbf{v} = \lambda_n(W)\mathbf{v}$, i.e. corresponding to the smallest eigenvalue of $W$, and $W\mathbf{u} = \lambda_2(W)\mathbf{u}$, i.e. corresponding to the second largest eigenvalue of $W$.

This is motivated by the observations in Lemma 13, where we noted that components in the eigenspace corresponding to the smallest eigenvalue of $W$ get amplified the most.

## C.2 Graph Topologies and Mixing Matrices

**Interpolated Ring (between uniform weights and interpolate with a fully-connected topology).** We consider the ring topology $W_{\text{ring}}$ on $n$ nodes, where each node $i$ has self weight $w_{ii} = \frac{1}{3}$ and $w_{i,1+(i \mod n)} = w_{i,(i-2 \mod n)+1} = \frac{1}{3}$ for its neighbors. We interpolate this uniform weight ring topology with a fully-connected topology, $W_{\text{complete}} = \frac{1}{n}\mathbf{1}\mathbf{1}^\top$, that is, $W_\alpha := \alpha W_{\text{ring}} + (1 - \alpha)W_{\text{complete}}$. The eigenvalues of $W_{\text{ring}}$ are $\lambda(W_{\text{ring}}) \in \left[-\frac{1}{3}, 1\right]$, and $\lambda(W_{\text{complete}}) \in [0, 1]$, and therefore $c$ of $W_\alpha$ is also a constant.

**Ring with smaller self weight.** We consider the ring topology $W_w$ on $n$ nodes, where each node $i$ has self weight $w_{ii} = w \leq \frac{1}{3}$ and $w_{i,1+(i \mod n)} = w_{i,1+(i-2 \mod n)} = \frac{1-w}{2}$ for its neighbors. The eigenvalues of $W_w$ are $\lambda(W_w) \in [2w - 1, 1]$, and therefore $c$ can become small by choosing $w$ (note that the $\lambda_n(W_w)$, while decreasing for smaller $w$, is not equal to $2w - 1$ in general, expect when $w = \frac{1}{3}$). We measure the exact value $\lambda_n(W_w)$ when reporting $c$ below.

## C.3 Additional Plots for Setup A

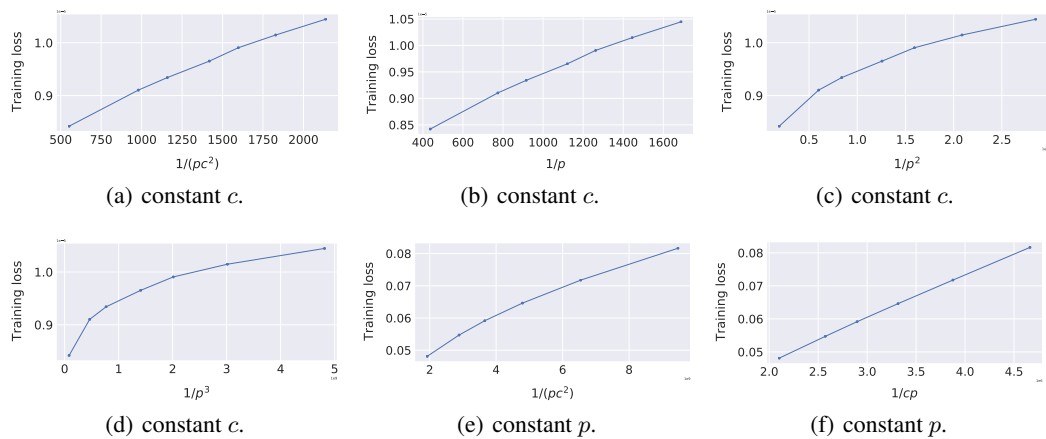

(a) constant $c$.      (b) constant $c$.      (c) constant $c$.

(d) constant $c$.      (e) constant $p$.      (f) constant $p$.

Figure 4: Impact of $c$ and $p$ on the convergence with the Gaussian stochastic noise $\sigma^2 = 1$. The first four subfigures illustrate the impact of $p$ on convergence when $c$ is kept constant; showing a linear scaling of the loss compared to $\frac{1}{p}$. The last subfigure varies $c$ in the graph while keeping $p$ as a constant, and we see a linear scaling compared to $\frac{1}{c^2}$.

## C.4 Additional Plots for Setup B

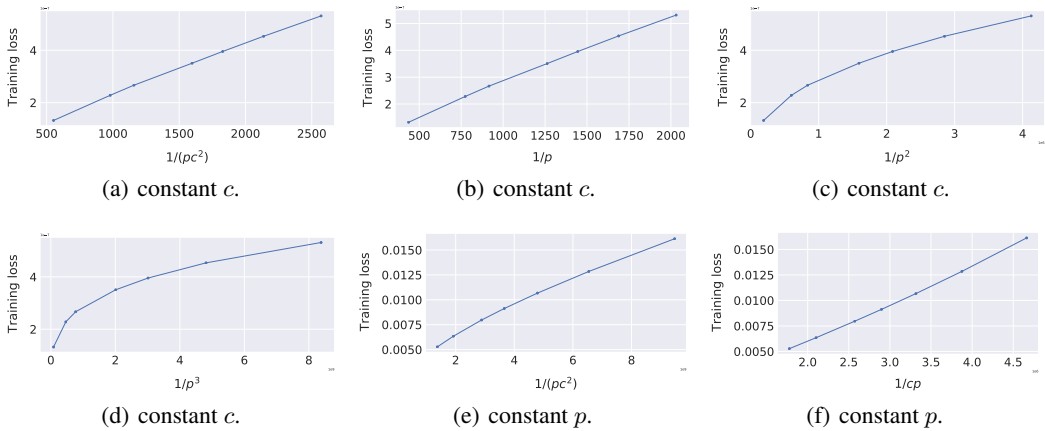

(a) constant $c$.      (b) constant $c$.      (c) constant $c$.

(d) constant $c$.      (e) constant $p$.      (f) constant $p$.

Figure 5: Impact of $c$ and $p$ on convergence with the structured stochastic noise $\sigma^2 = 1$. The first four subfigures illustrate the impact of $p$ on convergence when $c$ is kept constant; showing a linear scaling of the loss compared to $\frac{1}{p}$. The last subfigure varies $c$ in the graph while keeping $p$ as a constant, and we can see a linear scaling compared to $\frac{1}{c^2}$.

In Figures 4 and 5 we study the impact of $c$ and $p$ on the convergence. These findings support the $\mathcal{O}\left(\frac{\gamma^2 \sigma^2}{pc^2}\right)$ scaling predicted by theory—however, cannot replace a formal proof. We leave this for future work.