# OpenReview forum: "An Improved Analysis of Gradient Tracking for Decentralized Machine Learning"
_NeurIPS.cc/2021/Conference — NeurIPS 2021 Poster_

### Official Review · Reviewer_caUZ · 2021-07-09

**Rating:** 6
**Confidence:** 4

**Summary:**

This paper analyzes Gradient Tracking, a very common decentralized optimization algorithm, in the stochastic setting. The authors first introduce the standard gradient tracking algorithms, along with the assumptions they use to analyze it. Then, they provide convergence results in 3 main settings: non-convex, strongly-convex and weakly-convex. In each case, the leading order term matches the rates of centralized mini-batch SGD, meaning that the methods are fast (at least when this term dominates). It is argued that many interesting extensions naturally fit in this framework and could be adapted without too much efforts.

Then, a proof sketch is given, and highlights the importance of studying the consensus difference X_t - \bar{X_t}, where \bar{X_t} is the average over all nodes, instead of studying just X_t. The other key ingredient is that a simple recursion, can be written with the consensus difference on both GT variables. Although the operator J involved in this recursion is not contractive at each step (spectral radius greater than 1), J^i with i large enough is contractive (spectral radius smaller than 1/2), which allows to prove convergence of the recursion.

Then, the rates obtained for Gradient Tracking are compared with rates previously obtained in the literature, and they compare favorably overall. Finally, some toy experiments are given to analyze the impact of the graph-dependent constants c and p that were introduced and that appear in the higher order terms. These experiments seem to confirm the theoretical rates.

**Limitations And Societal Impact:**

No foreseeable societal impact

**Main Review:**

The paper is globally clear and well-written, and state of the art is reviewed well. The results are significant since they show that gradient tracking, a very standard approach for linearly converging decentralized learning, achieves good rates in the stochastic setting as well.

Reference [14] achieves similar results with a simple decentralized SGD algorithm. Yet, I believe that this work is interesting because compared to [14], this paper gets rid of a "data-heterogeneity" term. This is quite important and I believe it is precisely due to using gradient tracking (or any other method that would converge to the true global minimizer without stochasticity, which is not the case of D-SGD).

Time varying and local updates extensions: does it really only requires slight adaptations of the proof? If so, I believe this could significantly strengthen the results in the paper and should be included.

I believe that p should be explained more in details, since it is non-standard. In particular, it would be helpful to discuss its relationship with the spectral gap of the graph: I would say from Eq (3) that it is always bigger, but of the same order of magnitude since the max of the eigenvalues should be very close to 1, but it would be good to state this explicitly. Similarly, examples on some standard graphs (ring, grid) could be given. It would also be good to do the same for parameter c, although this is partially done in Section 6.1 (but it could be useful to have it earlier).

About Section 4.2: This seems to confirm the fact that p and delta are not so different actually. Indeed, in this case, gradient tracking converges similarly to standard (synchronous) gossip, but with delta replaced by p. Section 4.2 may be condensed into a remark to make room for more discussions on p and c.

Experimental evaluation of the dependence on p and c is a good idea. I believe that linear scaling in 1/p is quite natural (if it is closely related with the spectral gap). For the c^2 term, the range of parameters considered is a bit narrow, but since it is 1/c^2 I understand it is hard to considered much wider choices of parameters.

**Time Spent Reviewing:**

2

---

> ### Author Response · Authors · 2021-08-10
> **Reply to reviewer caUZ**
>
> Dear Reviewer, thank you for your suggestions to improve the paper.
>
> 1. As argued in the reply to reviewer TxXs, local updates are easy to add as the update matrix $W_{\tau,l}$ is symmetric and fixed (see Assumption 4 in [15]). We will add it to the next version.
>
>     To extend our proof to the general time-varying case is more difficult (due to non-symmetricity) and would require additional analysis. Please also check our reply to reviewer TxXs for more details.
>
> 2.  For a fixed matrix W, the exact relation between $p$ and spectral gap $\delta$ is $p = 2\delta - \delta^2$. From this we can conclude that $\delta \leq p \leq 2\delta$. Moreover, asymptotically as $\delta \to 0$, $p \approx 2\delta$. We will add this discussion to the paper.
>
>     We will also add a table of parameters $p$ & $c$ for some of the well-known graphs:
> |Topology |   p    |     c|
> |-------------|--------|--------|
> |Ring-n     | $\mathcal{O} ( 1/n^2 )$   |  1 -  (⅓)^2 = 8/9|
> |Torus-n    | $\mathcal{O} (1/n)    $    |  >= 4/5 (Remark 7) |
> |fully-connected | $\mathcal{O}(1)$  |   1 |
>
>     Note that all the $c$ values are constant for these topologies.

---

> > ### Comment · Reviewer_caUZ · 2021-08-24
> > **Thank you for your answer**
> >
> > I would like to thank the authors for their answer.
> >
> > I am not too bothered by the lack of experiments since GT is quite a standard algorithm, and I am not sure more involved experiments (e.g., deep learning) would bring something to a paper that is mainly about improving an analysis.
> >
> > I agree with other reviewers that the contribution is a bit incremental overall, but at the same time it has the benefit of obtaining clear improved rate, especially in the strongly convex case (which I find quite important).
> >
> > Besides, it seems to adapt nicely to local updates indeed, and I think that this should be clearly written and highlighted.
> >
> > I thus keep my score unchanged.

---

### Official Review · Reviewer_hLSN · 2021-07-13

**Rating:** 4
**Confidence:** 5

**Summary:**

The paper studies  the gradient tracking (GT) algorithm for stochastic distributed optimization problems over undirected, static graphs; nonconvex, strongly convex and weakly convex local objective functions are considered. The claimed contribution is a new line of analysis unlocking tighter (asymptotic) convergence rates than those already developed in the literature. The impact of some network parameters on the convergence rate is also discussed, and validated via some (albeit limited)  numerical results.

**Ethical Concerns:**

No ethical concerns.

**Limitations And Societal Impact:**

Some limitation in the algorithmic analysis and results are already mentioned in points 1 and 2 and 4 of my review above. I do not see other limitations.

**Main Review:**

The paper is overall well written (I found only minor typos); statement are rigorous and clear.

On the technical side, I think that the major differences from existing proofs (in particular those in [15]) are twofold:
1. Eq.  (11) and Lemma 5 open the way to establish contraction of the error dynamics; this approach is new to my knowledge, and is of general interest.
2. A thither analysis of the tracking error with respect to [15] is provided, which allows the authors to remove in the final error bound the dependence of data dissimilarity
However, the rest of the analysis is fairly standard and does not add much to the classical roadmap followed to study such distributed algorithms. Overall the technical contribution is a bit limited.

When it comes to the new rate bounds, I have some comments which deflect from authors' assessments.

1. A key motivation provided by the authors to study GT methods is the potential heterogeneity of data across the agents. But if this is the case, the assumption that all the functions f_i have the same strong convexity constant \mu and Liptschitz gradient constant L is quite restrictive. On the other hand, if such constants are different for each agent (which in my opinion is more compatible  with data heterogeneity) , then the convergence results would read in terms of L=\max_i L and \mu=\min_i \mu_i. If so, the claimed bound are no longer tight, as claimed by the authors. For instance, the following papers (although in the deterministic setting) provide tighter bounds for the strongly convex case (Table 1):

Haishan Ye,  Ziang Zhou,  Luo Luo, Tong Zhang, "Decentralized Accelerated Proximal Gradient Descent," NeurIPS 2020
(the non acceleration comes as special case, note the dependence of the rate on the global condition number)

Ying Sun, Amir Daneshmand, Gesualdo Scutari, Distributed Optimization Based on Gradient-tracking Revisited: Enhancing Convergence Rate via Surrogation," arXiv:1905.02637

including other non GT schemes, e.g.,

H Li and Z Lin "Revisiting extra for smooth distributed optimization," - SIAM Journal on Optimization, 2020

I think that a much stronger contribution would be developing the analysis under different \mu_i and L_i, still showing dependence of the rates on the parameters \mu and L of the average loss F. However, it is not clear to me that the current analysis extend to this more general case.

2. The comparison in the table can be improved. I do not think that all the rate bounds therein refers to the "same" optimality measure. Speaking of which, I think that characterizing the optimality of the average of the iterates does not capture directly the consensus error. I find more suitable a merit function in terms of the local agents' variables rather than just the average (which does not provide any info about the distance of the local copies from the solution). I encourage the authors to redo the analysis incorporating explicitly in the optimality matrix the consensus error.

3. The authors may want to compare their result with those in

K Yuan, S A. Alghunaim, B Ying and A H. Sayed "On the Influence of Bias-Correction on DistributedStochastic Optimization," IEEE Trans. Signal Processing, 2020

The exact diffusion scheme there is another form of gradient correction.

4. Simulations are limited. I would have expected a comparison of the GT with other distributed method employing some form of gradient correction to check whether these methods exhibit different behaviors in practice. This would also validate the tightness of the rate results.


Some typos:
------------------

1. line 113: Hence *and*

2. line 141: Section 4.1, *and .*



**Time Spent Reviewing:**

3

---

> ### Author Response · Authors · 2021-08-10
> **Reply to reviewer hLSN**
>
> As acknowledged by the reviewers TxXs and caUZ on the novelty and originality of this submission, we kindly disagree with the reviewer that technical contribution is limited. We believe that improving dependence on the graph parameter p is non trivial and cannot be seen as a simple combination of previous results. While the general proof sketch might look similar to the one in [15], looking at decrease with several steps is non-standard if the mixing matrix W doesn’t change. The result of key Lemma 3 is non obvious. It is also non-standard in the literature to analyse non-contractive operators J.
>
>
> 1. GT method has a vast amount of literature under the same assumptions on Lipschitz constant / strong convexity parameters as we imposed (see [22, 33, 34, 47, 53, 23, 16, 43]). These assumptions are also widely used for other methods: [15, 20, 48, 40] and many others. We believe that improving the convergence rate even under assumptions we used (the same local Lipschitz constants $L_i = L$ and strong convexity parameters $\mu_i = \mu$) is interesting and significant itself.
>
>     We agree that for heterogeneous functions it is interesting to consider the case with different $L_i$ and $\mu_i$ on different nodes, but this is orthogonal to our work and we leave it for future extensions.
>
> 2. Thanks a lot for your comment. We will improve presentation in Table 1 by adding the convergence metric.
>
>     We note that with our proof it is possible to obtain similar convergence bounds as in Theorem 1 for the averaged loss computed at local variables, i.e. $\frac{1}{n} \sum_i f(x_i)$. This is because our analysis bounds consensus distance $||\Psi||_F^2$ in Lemma 6. We will add this as a remark to the next version of the paper.
>
> 3. We note that exact diffusion is the same algorithm as $D^2$ (see [49]). We compare our results to $D^2$ [49] in section 6.3, and comment on line 71. We will also make it more clear in section 6.3.
>
> 4. The main contributions of this submission lie in the significance and novelty of improving the convergence rate on the graph parameter in the *non-leading* terms in all cases. Providing a thorough numerical comparison is beyond the scope of the paper, as many prior works [R2, 23, 33, 51] have already included the detailed comparisons.
>
>     However, we agree with the reviewer that including a set of illustrative experiments would further strengthen this work. We will include similar results (compare GT with diffusion, exact-diffusion/$D^2$), e.g. in [R2] (Figure 3 & 6 & 8), to investigate the impact of network topology; We will control the c & p derived from our analysis to manipulate the network topology to better identify the performance differences.
>
>     Please note that this is a minor addition, which was done in prior work already, that we are happy to provide.
>
>
> [R2] K Yuan, S A. Alghunaim, B Ying and A H. Sayed "On the Influence of Bias-Correction on DistributedStochastic Optimization," IEEE Trans. Signal Processing, 2020.

---

> > ### Comment · Reviewer_hLSN · 2021-08-22
> > **Reply to the Rebuttal**
> >
> > I wish to thank the authors for replying to my concerns/comments. However, I found their arguments quite weak (basically they do not provide a technical justification to confute my statements just replied with another statement). Hence, I confirm my rate (4).
> >
> > 1. I still believe that beyond (11) and Lemma 5, the technical novelty of the paper is incremental; as also noted by other reviewers, the rest of the analysis is pretty standard. The authors just replied that they disagree but they did not provide any technical argument in support of their statement.
> >
> > 2. The authors claimed that their new rate results provide a tight dependence on the problem/network parameters. Well, this is not the case for the optimization parameters, since they make the strong assumption that all the functions have the same strong convexity and Lipschitz constant of the gradient. When this is not the case, the dependence on the rate from L and $\mu$ will be in the form of $\max_i L_i$ and $\,in_i \mu_i$, which is very pessimistic and much worse than what obtained in the literature in many recent papers (the fact that in the past  people were making such assumption does not justify to keep doing so, especially if the contribution of the paper is not a new algorithm but a claimed tighter analysis). Hence, the results obtained by the authors in this regard are not tight at all. The answer "since other people did that, we do the same" does not address the concern.

---

> > > ### Author Response · Authors · 2021-08-30
> > > **counterexample (response to part 2)**
> > >
> > > Dear reviewer,
> > >
> > > We agree that it might look appealing to consider the case with different $L_i$ and $\mu_i$ on different nodes.
> > >
> > > However, below we present an example that shows that a complexity estimate depending on $O(L_{\rm max})$ is **best possible for GT in general, and we cannot expect a result for the general case that depends on $O(L_{\rm avg})$ only**. We therefore kindly ask the reviewer to reconsider his/her score, as his/her suggested direction will not yield better complexity results (unlike the reviewer speculated earlier).
> > >
> > > **Intuition**: Unlike in the distributed setting (with exact averaging on a parameter server or with all-reduce) in the decentralized setting with gossip averaging it is not straightforward to derive complexity results that depend only on the average of the $L_i$’s, instead of $\max_i L_i$. Intuitively, this is because information spreads only slowly in the graph and using too large stepsizes locally might lead to divergence.
> > >
> > > **Counter example**: We illustrate this in the following counterexample (the reviewer is encouraged to verify these claims numerically).
> > >
> > > Consider $n > 4$ nodes, connected on a ring topology with uniform averaging matrix (that is, $w_{i,j-1} = w_{i,i} = w_{i,j+1} = \frac{1}{3}$). For nodes $1 \leq i < n$ let $f_i(x)=\frac{1}{2}*x^2$ and $f_n=\frac{L_{\rm max}}{2} * x^2$ for $L_{\rm max} > 100$ (and $d=1$). It can be verified numerically, that starting from $x_0 = 1$, Gradient Tracking converges with the stepsize $1/L_{\rm max}$, however, **GT does not converge** with the stepsize $\frac{2}{\mathbf{n} L_{\rm avg}}$, with $L_{\rm avg} = \frac{n-1+L_{\rm max}}{n}$ (and does hence also not converge with the stepsize $\frac{1}{L_{\rm avg}}$).
> > >
> > > We acknowledge that this is not a formal argument, but the example highlights that there is additional difficulty (and conjectured impossibility) to derive convergence rates depending on $L_{\rm avg}$ instead of $L_{\rm max}$ in the general case.
> > >
> > >
> > > **Remark**: The reviewer points to [Haishan Ye, Ziang Zhou, Luo Luo, Tong Zhang, "Decentralized Accelerated Proximal Gradient Descent," NeurIPS 2020]. However, this paper does not consider the gradient tracking algorithm, and the studied algorithm requires multiple mixing steps every iteration, in contrast to the algorithm we study here.

---

> > > ### Author Response · Authors · 2021-08-30
> > > **technical arguments (response to part 1)**
> > >
> > > Dear reviewer,
> > >
> > > We strongly disagree with your comment on the technical novelty. Unfortunately, the reviewer does not give a reference for the referred 'standard results'.
> > >
> > > The closest work we could find, resembling the proof technique used in the proof of Theorem 2, is [[Nedić et al, SIAM J. Opt, 2017](https://arxiv.org/pdf/1607.03218.pdf)]. This paper also considers a contracting property after sufficiently many steps (their Lemma 2), however,
> > > - they do not consider the stochastic setting,
> > > - their complexity result depends on $O(\kappa^{1.5})$ vs. $O(\kappa)$ in our case,
> > > - their complexity result depends on $O(n^{4.5})$ vs. our result that does not depend on $n$.
> > >
> > > These are a few highlighted differences, that the techniques, and results, are vastly different of this prior work.

---

> > > > ### Comment · Reviewer_hLSN · 2021-08-30
> > > > **About local condition number and global**
> > > >
> > > > I think that I already provided several references of algorithms which are proved to converge linearly in the deterministic case, with a dependence better than $L_\max/\mu_\min$. I'm talking about deterministic problems and not stochastic, in response to the Referee who was claiming as key novelty of this work improving rates of this paper in the deterministic setting.
> > > >
> > > > - Haishan Ye, Ziang Zhou, Luo Luo, Tong Zhang, "Decentralized Accelerated Proximal Gradient Descent," NeurIPS 2020 (the non acceleration comes as special case, note the dependence of the rate on the global condition number)
> > > >
> > > > The comment of the authors that this scheme is not a GT is incorrect. If you consider unconstrained problems (like in this manuscript) and remove acceleration (no acceleration in this manuscript), this reduces exactly to the plain GT method. It is true that one may need multiple rounds of communications per iteration but this will affect the overall complexity by a log factor, which can be still better than having a dependence on local condition number as in this paper. This allows one to use larger step-size.
> > > >
> > > > - Ying Sun, Amir Daneshmand, Gesualdo Scutari, Distributed Optimization Based on Gradient-tracking Revisited: Enhancing Convergence Rate via Surrogation," arXiv:1905.02637v2
> > > >
> > > > This is a GT scheme whose convergence (up to log factors) depends on the global condition number and not the local one (on top of the fact that they can handle nonsmooth terms and constraints).
> > > >
> > > > Example 1 therein shows a toy problem where the local condition number is much larger than the global one (it can be arbitrarily larger). If applied to such a problem, the rate result of this manuscript would be vacuous, in contrast with the one obtained in Sun's paper, which depends on the global condition number.
> > > > Indirectly, this also comments the "counter-example" provided by the authors in the other reply, which does not formally disprove anything.
> > > >
> > > > H Li and Z Lin "Revisiting extra for smooth distributed optimization," - SIAM Journal on Optimization, 2020
> > > >
> > > > This is just to show that there are also other distributed algorithms out there with a better dependence on the condition number (average $\mu$ rather than $\min \mu_i$ ). I think that keep considering only local condition number is a bit anachronist.

---

> > > > > ### Author Response · Authors · 2021-08-30
> > > > > **joint effect of mixing and (local/global) condition number**
> > > > >
> > > > > Dear reviewer,
> > > > >
> > > > > > Haishan Ye, Ziang Zhou, Luo Luo, Tong Zhang, "Decentralized Accelerated Proximal Gradient Descent," NeurIPS 2020 (the non acceleration comes as special case, note the dependence of the rate on the global condition number)
> > > > >
> > > > > For their main theorem (even in the non accelerated case) to hold, the number of gossip steps per iteration needs to be sufficiently large. In the notation of our paper, this is equivalent to additionally imposing a **non-trivial lower bound on $p$**.
> > > > >
> > > > > > Ying Sun, Amir Daneshmand, Gesualdo Scutari, Distributed Optimization Based on Gradient-tracking Revisited: Enhancing Convergence Rate via Surrogation," arXiv:1905.02637v2
> > > > >
> > > > > The results of this paper (as highlighted in their Table 3) either require (i) a special topology (star), or (ii) sufficient mixing, either by 'exact averaging' through multiple gossip steps, or through an upper bound on $\rho$ (equivalent to a **non-trivial lower bound on $p$** in our notation).
> > > > >
> > > > > In summary, both of these works provide convergence results only on a limited set of mixing topologies, while we **do not** require a lower bound on $p$, i.e. our results hold for arbitrary mixing topologies.
> > > > >
> > > > > Please note that the setting of repeated gossip steps is covered in our paper (simply by considering the mixing matrix $W' = W^K$, where $K$ denotes the number of steps) but our results are **not limited** to this case alone.
> > > > >
> > > > > >  Indirectly, this also comments the "counter-example" provided by the authors in the other reply,
> > > > >
> > > > > Please note that for a cycle graph $p=\Theta(n^{-2})$, and therefore $p$ can be arbitrarily small. Hence, Theorem in [Sun et al.] becomes vacuous  for $n$ sufficiently large and does not cover our example.
> > > > >
> > > > > > I think that keep considering only local condition number is a bit anachronist.
> > > > >
> > > > > We respectfully disagree with this opinion. In our paper we advanced the state of the art even for the ‘anachronist’ setting.
> > > > >
> > > > > However, this discussion with the reviewer has revealed that there is an intimate connection between the global condition number and the mixing parameter $p$:
> > > > >
> > > > > - if $p$ is large (the setting in [Sun et al]), then the dependency on the local condition number can be improved to a dependency on the global condition number
> > > > > - on the other hand, if $p$ is arbitrarily small, our counterexample shows that the local condition number can in general not be replaced with the global condition number
> > > > >
> > > > > This hints to an interesting open problem. Perhaps one might conjecture that the complexity does not depend on $p$ and the condition number alone, but on the joint effect of both. We will add a discussion of discovery to the paper, and will gladly give credit to the anonymous reviewer for pointing this out.

---

> > > > > > ### Comment · Reviewer_hLSN · 2021-08-30
> > > > > > **Reply**
> > > > > >
> > > > > > Thanks to the authors for the interesting discussion and comments.
> > > > > >
> > > > > > As I already mentioned, the multiple rounds of communications in the papers above (which are necessary when $\tho$ is not sufficiently small) will translate in a log factor (independent on \epsilon but $L$) on the final complexity analysis. You do not need any condition on $\rho$ in that case. A log factor in a rate dependent on the global condition number can still provide better rate estimates than a rate expression dependent on the local condition number and no extra log factors (when the local condition number is much smaller than the global one). That was the sense of my previous comments.

---

> > > > > > > ### Author Response · Authors · 2021-08-30
> > > > > > > **Log factor**
> > > > > > >
> > > > > > > Dear reviewer,
> > > > > > >
> > > > > > > > A log factor in a rate dependent on the global condition number can still provide better rate estimates than a rate expression dependent on the local condition number and no extra log factors (when the local condition number is much smaller than the global one).
> > > > > > >
> > > > > > > Yes, we do not disagree with this comment.
> > > > > > >
> > > > > > > The reviewer points towards an interesting direction of future work that might improve our understanding of the performance of decentralized optimization schemes and the interplay of parameters that have been considered mostly in isolation so far in the literature (the comment of the reviewer does not seem to be limited to tracking algorithms alone, and could also yield more refined estimates for e.g. decentralized-SGD or other related schemes).
> > > > > > >
> > > > > > > We believe that the reviewer's main concern (that more attention should be devoted to refined smoothness assumptions in future work) can be addressed for the camera ready version by adding a discussion of the (sub)optimality of our results under refined smoothness assumptions. We will also cite the exemplary references indicated by the reviewer.

---

### Official Review · Reviewer_TxXs · 2021-07-14

**Rating:** 8
**Confidence:** 4

**Summary:**

This paper studies the gradient tracking method for decentralized optimization. While it is widely used, its convergence rate is not optimal compared with some other methods. This paper provides a tighter analysis for nonstrongly convex problems, strongly convex problems, and nonconvex problems. Faster convergence rates are proved in this paper.

**Limitations And Societal Impact:**

I suggest to compare with "Sulaiman A. Alghunaim, Ernest K. Ryu, Kun Yuan, and Ali H.Sayed. Decentralized proximal gradient algorithms with linear covnergence rates. IEEE trans. on automatic control", and discuss the complexity comparisons.

**Main Review:**

Originality: This paper studies a widely used method and gives improved convergence rates analysis. The analysis framework follows [15]. However, a few crucial differences remain due to the specification of GT, and moreover, improved convergence rates are proved. To my best knowledge, all the previous conplexities depend on $\frac{1}{p^2}$, while this paper improves it to $\frac{1}{pc}$, and $c$ can be controled in practice, such as $c=1$ when we set $W$ to be $\frac{I+W}{2}$. So I think the analysis in this paper is novel.

Quality: The theory is technically solid. This is a complete work. However, I think more details of the proof should be provideded in the final version.

Clarity: I think this paper should be written more carefully in the final version.

Significance: I think the results in this paper are important. It provides a new way to analyze GT, and faster convergence rates are proved than previous work.

Comments:

1. The authors claim that they improve over all known results on line 41. I suggest to cite "Sulaiman A. Alghunaim, Ernest K. Ryu, Kun Yuan, and Ali H.Sayed. Decentralized proximal gradient algorithms with linear covnergence rates. IEEE trans. on automatic control", which proved the $O( ( \frac{L}{\mu} + \frac{1}{p^2} )log\frac{1}{\epsilon} )$ complexity for strongly convex problems. As a comparison, this paper only proves the $O( \frac{L}{\mu p} log\frac{1}{\epsilon} )$ one ($\sigma=0$ and $c=1$ in (7)), which is slower when $\frac{L}{\mu} > \frac{1}{p}$.

2. The GT algorithm on line 112 does not lead to steps 3 and 5 in Algorithm 1. I suggest to write W on the left hand side.

3. What is $\eta_t$ in step 3 of Algorithm 1? The stepsize is $\gamma$ in the input description.

4. The authors claim that their results can be used to time-varying graphs on line 172. I am not sure whether it is correct because there are some challenges, such as the non-symmetry of $W_{l,\tau}$ (thus, no eigenvalue decomposition exists), which is defined in Assumption 4 in [15].  Can the convergence rates still be improved over time-varying graphs? Consider (7) for example, I am interested whether the $O( \frac{\sigma^2}{\mu n T} + \frac{L\sigma^2\tau}{\mu^2 p c^2 T^2} + \frac{LR_0\tau}{pc}\exp(-\frac{\mu pcT}{L}) )$ complexity can be proved, where  $\tau$ is the parameter in Assumption 4 in [15].

5. What is $\widetilde W$ on line 189? Is should be defiend in Section 5, rather than in Supp. A.

6. What is $0<a11$ on line 555?

7. How the inequality below line 571 comes? I think more details should be provided, rather than checking the proofs in [15].




**Time Spent Reviewing:**

more than 24 hours

---

> ### Author Response · Authors · 2021-08-10
> **Reply to reviewer TxXs**
>
> Dear Reviewer,
>
> We would like to thank you for your time spent reviewing our paper and for providing detailed comments.
>
> 1. Thanks a lot for the reference. We will include it in the next version.
>
> 2,3,5,6. Thanks for pointing out these typos. We will fix it in the next version. On line 555 it was supposed to be  0 < a < 1
>
> 4. Thanks for your comment. We realized that it is not straightforward to extend our result for the general time-varying case. The matrix $W_{l, \tau}$ (see Assumption 4, [15] for definition) is not necessarily symmetric, and might change with time, therefore the eigenvalue decomposition cannot be applied. A key technical challenge is hidden in Lemma 11. It is not trivial to bound the quantity of Lemma 11 by a constant, and would be an interesting question for future work. We will adjust our claims regarding the time-varying graph extension in the paper and will point out these challenges.
>
>      However, please note that there are some practical examples for which *it is possible to extend our result to certain time-varying topologies*.  For example, for *decentralized SGD with local steps*, the mixing matrix $W_{l, \tau}$ is equal to W and therefore is symmetric and does not change across rounds. In this case we can define p and c similarly as in our Assumption 1, to be the parameters of the matrix $W_{l, \tau} \equiv W$. This analysis will lead to a convergence rate $O\left( \frac{\sigma^2}{\mu n T} + \frac{L \sigma^2 \tau}{\mu^2 p c^2 T^2} + \frac{L R_0 \tau}{pc} \exp\left( - \frac{\mu p c T}{L} \right)  \right)$ in the strongly convex case. We will add this extension to the updated version of the paper.
>
> 7. Thanks for this comment. The inequality below line 571 follows from the fact that $||J^i||^2 \leq 1/p~~ \forall i$ (the proof is similar to Lemmas 11,12, given (11) ). The rest of the proof consists of unrolling these inequalities as in [15]. We agree that more detailed proof will greatly increase readability.
> In fact, we found an even simpler way to finish the proofs that significantly simplifies the approach in [15] (and our current proof). That is, by including the residual sum $E_{t + \tau} = \frac{1}{\tau} \sum_{j = 0}^{\tau - 1} \Theta_{t + \tau + j} $ in the Lyapunov function.
>
>     We will include detailed derivations in the next versions and make all the proofs self-contained.

---

> > ### Comment · Reviewer_TxXs · 2021-08-20
> > **Response to the rebuttal**
> >
> > Thanks for the rebuttal. I keep my score unchanged.
> >
> > By the way, for the 4th response, I wonder whether the order of $\tau$ will increase in the case with local steps. I suggest the authors to prove the complexity carefully before adding this extension.

---

### Official Review · Reviewer_x8WX · 2021-07-16

**Rating:** 4
**Confidence:** 5

**Summary:**

This work studies the problem of decentralized nonconvex optimization problems and it is relevant to the conference. The paper provides convergence analysis for an existing and well-studied algorithm -- stochastic gradient tracking algorithms, without providing any improvements. The paper prove its algorithm the same convergence compared to existing works. There are some light and limited experiments associated with the paper, but only limited to quadratic case.

**Limitations And Societal Impact:**

Please provide discussion on the limitations and potential negative societal impact of the work per NeurIPS' request.

**Main Review:**

The main contribution of the paper is the offering of convergence analysis for an existing and well-studied algorithm -- stochastic gradient tracking algorithms for decentralized nonconvex optimization. The problem is well-motivated and the writing is clear. There are some light simulation results thus the proposed algorithm seems work from both the theory and simulations.

The gradient tracking algorithm for decentralized nonconvex optimization is well-studied by [1], and the linear speed up rate is also well-explored by many scholars [2]. The main contribution of this work is a combination of above works into a new paper, without any convergence rate improvement [both at $\sigma / \sqrt{nT}$]. Also, the simulation is based on simple quadratic functions only, which is way too simple in current deep learning era and further weaken the work significance.

In short, the derived algorithm and rates seems to be correct. However, the paper is still a combine-then-twist work based on top of several existing works and show no improvement compared to existing works. The paper could be strengthened by demonstrating more significant results instead of incremental.

[1] P. Di Lorenzo and G. Scutari. Next: In-network nonconvex optimization. IEEE Transactions on Signal and Information Processing over Networks, 2(2):120–136, 2016.

[2] Hanlin Tang, Xiangru Lian, Ming Yan, Ce Zhang, and Ji Liu. "$ D^ 2$: Decentralized training over decentralized data." In International Conference on Machine Learning, pp. 4848-4856. PMLR, 2018.

============================================
After response period: I have carefully and thoroughly read the article and all responses and discussions. As many other reviewers also mentioned, this work is trivial and includes many typos and flaws. I keep my score unchanged.

**Time Spent Reviewing:**

5

---

> ### Author Response · Authors · 2021-08-10
> **Reply to reviewer x8WX**
>
> We kindly disagree with the reviewer’s statement that the paper “doesn’t provide any improvements” compared to existing works.
> In contrast to what is claimed by the reviewer, our analysis improves the dependence of convergence rate on the graph parameter p in all the cases (convex, strongly convex, non-convex) with respect to prior work (see Tables 1, 2). We believe this is a non-trivial / non-incremental contribution.
>
> Hence, we would like to urge the reviewer to reconsider their review and adjust their score.
>
>
>
>
> The reviewer hinges his/her criticism only on the statistical term in the rate that does not depend on the graph parameter. It is obvious that this term cannot be improved: firstly, because it does not depend on p, and secondly, because the term is known to be optimal [see R1] for stochastic algorithms in general.
>
> We agree with the reviewer that for the gradient tracking algorithm the leading stochastic noise term in the strongly-convex case (\sigma^2/nT) was derived in earlier works already, for instance in the cited work [33].
>
> We discuss technical limitations of our framework in Sections 6 and 8. As we neither propose new algorithms nor new applications, we do not foresee negative societal impact from our theoretical contribution.
>
> [R1] Nemirovsky, A. S. and Yudin, D. B. Problem complexity and method efficiency in optimization. Wiley, 1983.

---

### Author Response · Authors · 2021-08-10
**Reply to all reviewers**

We would like to thank the reviewers for their valuable comments that help to improve our manuscript. We addressed comments from each of the reviewers separately below each of the reviews.

---

### Decision · Program_Chairs · 2021-09-27

**Decision:**

Accept (Poster)

**Comment:**

This paper studies gradient tracking for stochastic decentralized learning. Their new analysis provides improved convergence rates for strongly convex functions. In particular the dependence on p (which is essentially the spectral gap) is changed to pc^2, which is always better since c>p. Moreover, as noted in the paper, c can be controlled and therefore can provide better guarantees in practice as well. This is a clean observation, and perhaps of broader interest if (11) and Lemma 5 can be used to prove contraction in other problems.

With multiple rounds of communications a reviewer has noted that better rakes in terms of k_g can be obtained. It would be good to include that remark, and in particular emphasize that this paper does not consider multiple rounds of communication. The authors' claim about the analysis extending to time varying graphs and local updates easily are disappointing. Primarily because it is not true as one of the reviewer points out (which the authors agree to), and partly because this is a sloppiness where I am not sure if the authors even analyzed it themselves before putting this claim in the paper. At a latter point someone may think about obviousness of these things when they are unable to extend the results.